# SWE-Compass: Towards Unified Evaluation of Agentic Coding Abilities for Large Language Models

Jingxuan Xu [*1]  Ken Deng [*1]  Weihao Li [*1]  Songwei Yu [*1]  Haoyang Huang [*1]  Xinping Lei [1]  Yifan Yao [*1]
Huaixi Tang [*1]  Zhiyi Lai [*1]  Kepeng Lei [*1]  Zizheng Zhan [*1]  Yanan Wu [*2]  Chenchen Zhang [*3]
Wenqiang Zhu [1]  Wen Xiang [1]  Zongxian Feng [1]  Han Li [3]  Junqi Xiong [3]  Dailin Li [1]  Zuchen Gao [1]  Kun Wu [1]
Yuanxing Zhang [1]  Wuxuan Gong [1]  Ziyuan Gao [1]  Guanxiang Wang [1]  Yirong Xue [1]  Mengfei Xie [1]
Xiaojiang Zhang [1]  Jinghui Wang [1]  Wenhao Zhuang [1]  Zheng Lin [1]  Huiming Wang [1]  Zhaoxiang Zhang [1]
Yuqun Zhang [4]  Haotian Zhang [1]  Ming Sun [1]  Chen Bin [1]  Jiaheng Liu [3]

## Abstract

Evaluating large language models (LLMs) for software engineering has been limited by narrow task coverage, language bias, and insufficient alignment with real-world developer workflows. Existing benchmarks often focus on algorithmic problems or Python-centric bug fixing, leaving critical dimensions of software engineering underexplored. To address these gaps, we introduce **SWE-Compass**, a comprehensive benchmark that unifies heterogeneous code-related evaluations into a structured and production-aligned framework. SWE-Compass spans 8 task types, 8 programming scenarios, and 10 programming languages, with 2000 high-quality instances curated from authentic GitHub pull requests and refined through systematic filtering and validation. We benchmark ten state-of-the-art LLMs under two agentic frameworks, SWE-Agent and Claude Code, revealing a clear hierarchy of difficulty across task types, languages, and scenarios. Moreover, by aligning evaluation with real-world developer practices, we hope SWE-Compass can provide a rigorous and reproducible foundation for diagnosing and advancing agentic coding capabilities in large language models.

---

[*]Equal contribution  [1]Kuaishou Technology, Beijing, China  [2]Beijing University of Posts and Telecommunications, Beijing, China  [3]Nanjing University, Nanjing, China  [4]Southern University of Science and Technology, Shenzhen, China. Correspondence to: Jiaheng Liu <liujiaheng@nju.edu.cn>.

*Proceedings of the $43^{rd}$ International Conference on Machine Learning*, Seoul, South Korea. PMLR 306, 2026. Copyright 2026 by the author(s).

## 1. Introduction

Large language models (LLMs) trained on code have rapidly advanced from solving algorithmic puzzles to assisting with production-scale software development. Modern coding LLMs (Team et al., 2025b;c; Anthropic, 2025; Team et al., 2025a; Team, 2025) now exhibit strong multi-turn reasoning, long-context handling, and tool-use capabilities, enabling them to serve as autonomous coding agents that plan, edit, test, and deploy software. This shift has motivated a wave of benchmarks designed to measure their utility. However, existing evaluations fall short in capturing the full scope of real-world software engineering: most remain restricted to single-file tasks, Python-centric bug fixing, or synthetic algorithmic problems (Zheng et al., 2023b; Austin et al., 2021; Li et al., 2022; Jain et al., 2024; Zhuo et al., 2025), leaving critical developer activities such as feature implementation, refactoring, configuration, and performance optimization underexplored.

Recent repository-grounded benchmarks, such as SWE-bench and its variants (Jimenez et al., 2024; Yang et al., 2025c; Badertdinov et al., 2025; Yang et al., 2025c), have improved ecological validity by embedding evaluations in real issues, integrating test oracles, and introducing multi-language (Rashid et al., 2025; Yang et al., 2025c) or multimodal (Yang et al., 2025a) extensions. Yet these efforts largely converge on bug fixing as the dominant evaluation axis. As a result, they neglect the breadth of software engineering workflows that unfold across diverse scenarios—ranging from infrastructure and security engineering to machine learning system development—and across heterogeneous programming ecosystems. This narrowness prevents systematic capability diagnosis and obscures whether strong performance arises from generalizable reasoning or from artifact-specific adaptation.

To address these limitations, as shown in Figure 1, we present **SWE-Compass**, a unified benchmark comprising 2,000 verified instances for evaluating LLMs' agentic cod-

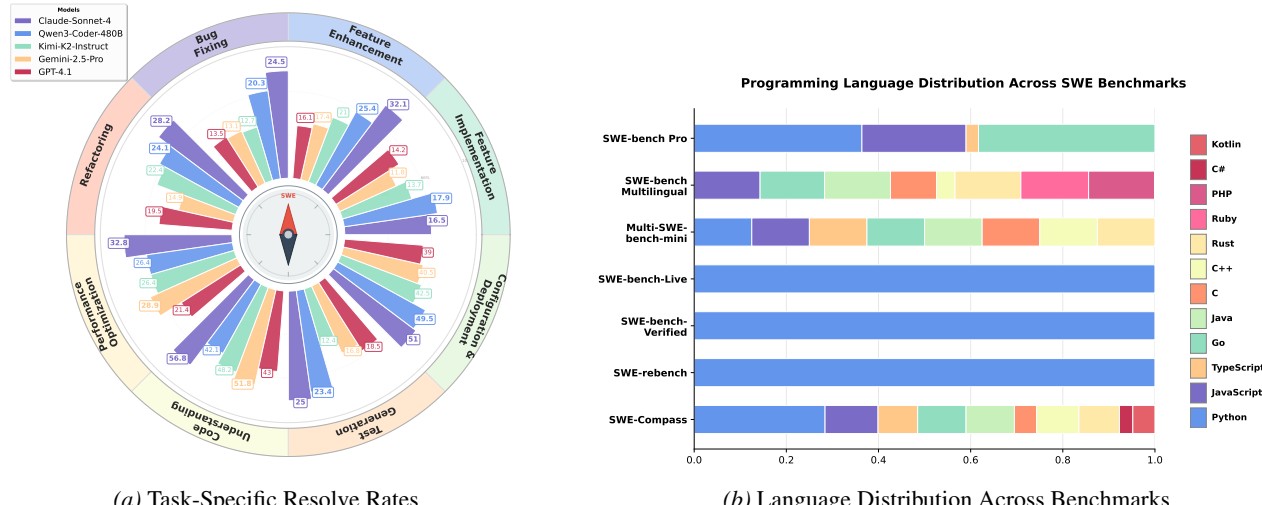

*(a)* Task-Specific Resolve Rates  *(b)* Language Distribution Across Benchmarks

*Figure 1.* Model performance across task types (left) and language coverage across benchmarks (right).

ing abilities. SWE-Compass spans 8 task types, 8 programming scenarios, and 10 programming languages, combining broad coverage with rigorous evaluation fidelity. Each instance is paired with executable environments and reproducible tests, enabling fair comparison across prompting and agent-based methods under controlled budgets. Importantly, SWE-Compass is built upon four design principles: (i) real-world alignment, ensuring data originates from genuine developer interactions; (ii) comprehensive coverage across diverse tasks and languages; (iii) systematic taxonomy, providing structured labeling and balanced distributions; and (iv) evaluation fidelity, guaranteeing that all instances are executable and verifiable. Together, these principles yield a benchmark that reflects the complexity, diversity, and reproducibility demanded by modern software engineering.

Our contributions are threefold:

- **A comprehensive, execution-grounded benchmark for software engineering.** We introduce SWE-Compass, a large-scale benchmark comprising 2,000 curated instances that span eight task types, eight programming scenarios, and ten programming languages. Each instance is drawn from real-world GitHub pull requests and paired with a reproducible execution environment, enabling rigorous and faithful evaluation of model performance in realistic development workflows.
- **A systematic evaluation framework aligned with real-world developer activities.** SWE-Compass establishes a structured taxonomy to assess models across different dimensions such as feature implementation, refactoring, test generation, and deployment. This design enables fine-grained diagnosis of LLM capabilities and provides a principled foundation for comparing agentic coding systems under consistent conditions.

- **Comprehensive empirical analysis and insights into LLM coding behavior.** Experiments with state-of-the-art LLMs and agentic systems reveal persistent gaps across tasks, languages, and scenarios, highlighting the difficulty of scaling beyond bug fixing and emphasizing the need for benchmarks that reflect the full complexity of real-world software engineering.

## 2. Related Works

**Coding LLMs and Agents.** Code large language models (Code LLMs) (Chen et al., 2021; Zhao et al., 2024; Chowdhery et al., 2023; Nijkamp et al., 2023; Fried et al., 2023; Xu et al., 2022; Roziere et al., 2023; Hui et al., 2024b; Deng et al., 2025; Que et al., 2024) — excel at a wide range of programming tasks, including code generation, completion, repair, translation, code comprehension, documentation generation, and cross-language migration, among others. Crucially, modern Code LLMs combine ultra-long context support with robust tool-calling capabilities, enabling them to maintain global awareness across large codebases while actively invoking editors, shells, debuggers, or web browsers (Liu et al., 2024c; 2025; Wang et al., 2024). This synergy has fueled the rise of agentic coding systems — such as SWE-Agents (Yang et al., 2024), OpenHands (Wang et al., 2025), and Claude Code (Anthropic, 2025), Qwen-Code (Team, 2025), Codex (OpenAI, 2025), Cline (Cline, 2024) — that autonomously plan, search, edit, test, and even perform agentic browser use to fetch live API documentation or solutions.

**Coding Benchmarks.** Single-file code benchmarks — such as HumanEval (Zheng et al., 2023b), MBPP (Austin et al., 2021), CodeContests (Li et al., 2022), LiveCodeBench (Jain et al., 2024) and BigCodeBench (Zhuo et al., 2025) — eval-

*Table 1.* Comprehensive comparison of SWE-Compass with existing benchmarks across different dimensions.

| Benchmark | # Samples | Lang | Task Types | # Repos | # Modified Files (Avg.) |
|---|---|---|---|---|---|
| HumanEval | 164 | Python | Algorithm | — | 1.0 |
| SWE-Bench-Verified | 500 | Python | Bug Fixing | 12 | 1.3 |
| SWE-Bench-Live | 1,319 | Python | Bug Fixing | 93 | 3.3 |
| SWE-Bench-Multilingual | 300 | 9 | Bug Fixing | 42 | 1.3 |
| Multi-SWE-Bench | 1,632 | 7 | Bug Fixing | 39 | 4.9 |
| SWE-Bench-Pro | 1,865 | 4 | 4 Types | 41 | 4.1 |
| **SWE-Compass (Ours)** | **2,000** | **10** | **8 Types** | **40** | **4.7** |

uate models on isolated algorithmic problems, abstracting away the structural, contextual, and environmental complexity inherent in real-world software engineering (Liu et al., 2024b); while SWE-bench (Jimenez et al., 2024) and its variants — including Multimodal SWE-bench (Yang et al., 2025b), SWE-bench Multilingual (Yang et al., 2025c), SWE-bench-Live (Zhang et al., 2025c), SWE-Lancer (Miserendino et al., 2025), SWE-rebench (Badertdinov et al., 2025) and others — have substantially improved ecological validity by grounding evaluation in real repository issues and incorporating dimensions such as visual context, multilanguage support, tool interaction, and repository-scale execution, they remain overwhelmingly confined to bug fixing as the de facto evaluation paradigm, neglecting the broader spectrum of developer activities. Details are provided in Table 1( Please see Appendix A.8 for more comparison with existing benchmarks). To address this gap, we introduce a benchmark that explicitly structures evaluation along orthogonal axes of task type and programming scenario.

## 3. SWE-Compass

### 3.1. Overview

Existing software engineering benchmarks primarily focus on Python-centric bug fixing tasks, exhibiting limited task coverage and insufficient alignment with real-world developer activities. In contrast, SWE-Compass is constructed from authentic software engineering requirements (Table 1), collecting high-quality repositories from GitHub pull requests through a multi-stage filtering process. The resulting benchmark encompasses 2000 instances across **8 task types**, **8 programming scenarios**, and **10 programming languages**, as shown in Figure 6 in Appendix A.2. It enables holistic evaluation of key capabilities, including bug fixing and performance optimization, in realistic software engineering contexts.

### 3.2. Design Principles

SWE-Compass follows four guiding principles:

- **Real-World Alignment**: We use authentic developer workflows from GitHub and Stack Overflow discussions, ensuring realistic and non-synthetic scenarios.

- **Comprehensive and Balanced Coverage**: We cover a broad spectrum of software engineering activities, including implementation, enhancement, maintenance, testing, and deployment, with balanced distributions across scenarios and 10 languages. In contrast to Python-centric bug-fixing benchmarks, SWE-Compass includes underrepresented tasks such as refactoring, performance optimization, and code understanding.

- **Systematic Taxonomy**: We employ an iterative active learning pipeline to derive a structured taxonomy of task types, scenarios, and languages. enabling scalable data collection.

- **Evaluation Fidelity**: All instances are tied to executable test patches and reproducible environments. For underrepresented tasks, we supplement data through controlled synthesis while maintaining realism.

### 3.3. Benchmark Construction

The construction of SWE-Compass follows a systematic and scalable approach organized into five major steps to ensure comprehensive coverage, balance, and real-world relevance: **(1) user analysis**, **(2) data collection**, **(3) environment building**, **(4) task construction**, and **(5) data validation**, as illustrated in Figure 2. Specifically, through an iterative *Active Learning* procedure applied to real-world coding conversations, we first identified that user needs predominantly fall into eight distinct task types, eight representative programming scenarios, and ten programming languages. We then collected a large volume of high-quality pull request (PR) data from GitHub repositories. By combining automated processing with expert annotation, we successfully built a set of executable development environments. Next, for each task type, we constructed and synthesized the corresponding task instances. Finally, after a multi-round filtering and quality validation process, we curated the SWE-Compass benchmark as the final dataset.

#### 3.3.1. STEP 1: USER ANALYSIS

To ensure that the evaluation accurately reflects model capabilities in realistic software development contexts, we collected repository-level coding discussions from two major platforms—**Stack Overflow** and **GitHub**. To discover

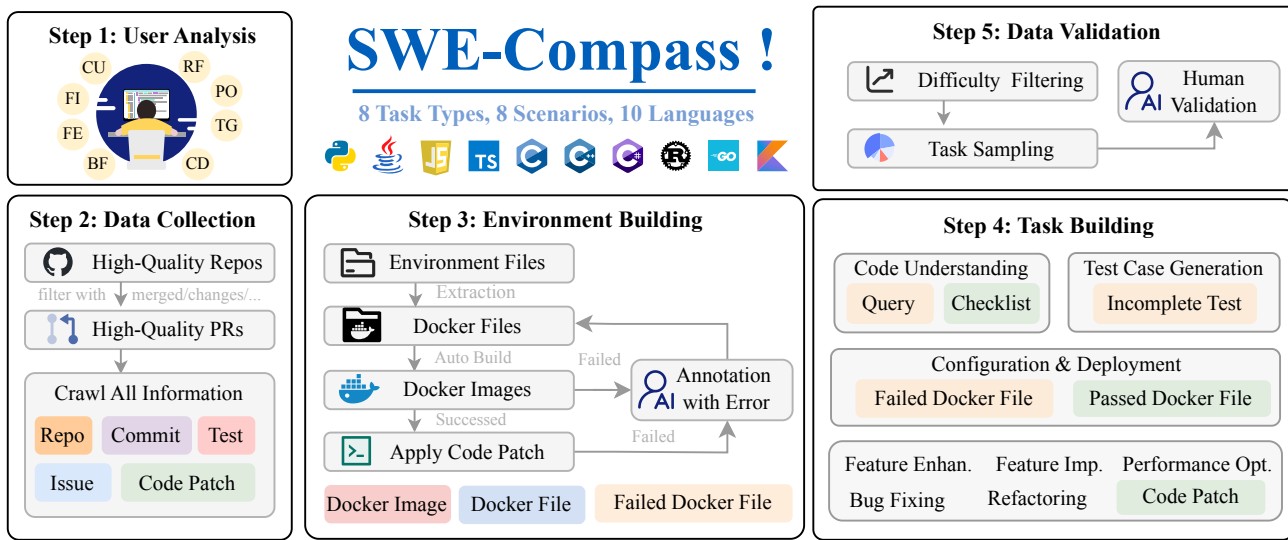

*Figure 2.* Construction of SWE-Compass.

emerging task categories, we designed an automated Active Learning framework for category discovery. Specifically, four popular software-related topics were chosen as initial label seeds for both task types and programming scenarios. Using an In-Context Learning (ICL)-based labeling approach, a large language model (LLM) was employed to annotate the collected conversations across three dimensions: task type, programming scenario, and programming language. Subsequently, tag clustering and LLM-guided seed optimization (via addition, modification, or deletion of tags) were applied to refine the label pool. The iterative process continued until convergence—when the updated seed pool no longer significantly differed from the previous ICL-generated pool. In our experiments, the `Qwen3-Coder-30B-A3B-Instruct`[1] model was used as the LLM annotator, and five iterations were performed in total. Ultimately, we identified **eight task types** (i.e., Feature Implementation (FI), Feature Enhancement (FE), Bug Fixing (BF), Refactoring (RF), Performance Optimization (PO), Code Understanding (CU), Test Case Generation (TG), Configuration & Deployment (CD)), **eight programming scenarios** (i.e., Application Development (AD), Database Systems (DS), Data Science & Engineering (DE), Machine Learning & AI (ML), Infrastructure Development (ID), Specialized Programming Domains (SPD), Security Engineering (SE), UI/UX Engineering (UI/UX)), and **ten major programming languages** (i.e., Python, JavaScript, TypeScript, Java, C, C++, Go, Rust, Kotlin, C#). Details are shown in Appendix A.2.

---

[1]https://huggingface.co/Qwen/Qwen3-Coder-30B-A3B-Instruct

### 3.3.2. STEP 2: DATA COLLECTION

To ensure both coverage and realism, we first collected existing open-source SWE benchmarks (e.g., Python bug-fixing datasets) and mapped them to our taxonomy of task types, programming scenarios, and languages. As shown in Appendix A.8, these benchmarks suffer from substantial limitations, including missing task types, highly imbalanced scenario distributions, and severe programming language skew. To mitigate these issues, we further augmented the dataset with high-quality repositories and PRs from GitHub.

- **High-Quality Repository Acquisition.** Repositories were filtered based on multiple quality criteria, including valid open-source licenses, at least 500 stars, active maintenance within the past six months, a minimum of three contributors, over 1,000 issues and PRs, more than 200 forks, and the availability of unit tests.
- **High-Quality PR Acquisition.** From the selected repositories, we extracted PRs and applied multi-stage filtering to retain those with clear modification semantics. Specifically, we kept PRs that were merged into the main branch, linked to descriptive Issues, and involved identifiable file- or line-level code changes. Each retained PR contained complete metadata, including issue, commits, test patches, code patches, etc.

After all filtering stages, approximately **50,000 high-quality PRs** were preserved, serving as the foundation for subsequent environment and task construction.

### 3.3.3. STEP 3: ENVIRONMENT BUILDING

To enable reproducible execution and evaluation, we constructed isolated containerized environments for all selected PRs. For each PR, environment dependencies (e.g., package managers, libraries, build tools, and runtime versions)

were automatically extracted from configuration files such as `requirements.txt`, `setup.py`, `Makefile`, and CI/CD scripts, and programmatically assembled into Dockerfiles to build initial images.

Each successfully built image was validated by running the repository's native test suite to ensure end-to-end executability and consistent behavior before and after patch application (F2P/P2P). Due to the complexity of real-world dependencies, the initial automated build success rate was approximately 2%. To recover failed builds, 30 expert annotators analyzed build logs, diagnosed root causes (e.g., missing dependencies or version conflicts), and applied targeted fixes before re-triggering builds on a Kubernetes cluster, increasing the overall retention rate to about 8%.

Finally, we obtained approximately **4,000 runnable Docker images**, each providing a fully reproducible execution environment for subsequent task synthesis and evaluation.

### 3.3.4. STEP 4: TASK BUILDING

Given the heterogeneity of the eight software engineering task types, we designed three complementary strategies: **Checklist Synthesis**, **Reverse Masking**, and **Heuristic Filtering**, each tailored to the characteristics of the corresponding tasks to balance realism and evaluation reliability.

**Checklist Synthesis.** For *Code Understanding*, we constructed instances from real PRs using `Issue`, `Code Patch`, and `Test Patch`. GPT-5 (Singh et al., 2025) was used to generate multiple natural-language queries per PR, followed by difficulty-aware filtering to remove trivial or ambiguous cases. For each retained query, GPT-5 further produced structured reasoning checklists covering functional intent, dependencies, and code effects, serving as verifiable anchors for LLM-as-a-Judge evaluation.

**Reverse Masking.** For deployment- and testing-related tasks, we started from verified "golden" artifacts and introduced controlled perturbations. For *Configuration & Deployment*, dependency packages in the `Dockerfile` were randomly removed or replaced, yielding **Failed Docker Files**; only cases with reproducible build failures or functional inconsistencies (P2F) were retained. For *Test Case Generation*, we selected PRs introducing more than five new test functions and constructed prompts from the corresponding `Code Patch` and `Test Patch`. Instances producing **Incomplete Tests** were kept to evaluate models' ability to generate complete and correct test suites, measured by correctness and coverage.

**Heuristic Filtering.** For patch-based tasks, we directly leveraged real-world PRs with targeted filtering. PRs passing all tests before and after patch application and achieving

over 30% runtime improvement were labeled as ***Performance Optimization*** seeds, with GPT-5 verifying alignment with performance-related issues. ***Refactoring*** instances were selected based on structural or readability improvements without behavioral changes. The remaining tasks were classified by patch intent: introducing new functionality (***Feature Implementation***), extending existing features (***Feature Enhancement***), or resolving errors and failing tests (***Bug Fixing***). All instances were validated for logical consistency and build reproducibility.

### 3.3.5. STEP 5: DATA VALIDATION

To ensure benchmark quality and diversity, we applied a structured validation pipeline that controls instance difficulty, balances task coverage, and ensures overall reliability.

- **Difficulty Filtering.** Candidate instances were screened based on the number of modified files, changed lines, and predictions from multiple model inferences, retaining samples with moderate and meaningful complexity.
- **Task-balanced Sampling.** We performed balanced sampling across task types and programming scenarios, with sampling weights adjusted to reflect realistic distributions over 10 programming languages.
- **Manual Verification.** All sampled instances were manually validated for executability, correctness, and semantic consistency across commits, queries, Docker images, and test cases.

The resulting benchmark, **SWE-Compass**, comprises **2,000 high-quality instances**, balanced across task categories, programming scenarios, and languages, and provides a reliable evaluation suite for assessing large language models in real-world software engineering settings.

### 3.4. Evaluation Metrics

We adopt task-specific evaluation metrics to assess model performance, with detailed definitions in Appendix A.3. Specifically, **Pass@1** is used for *Feature Implementation*, *Feature Enhancement*, *Bug Fixing*, and *Refactoring*. **Performance Optimization Score** is applied to *Performance Optimization*. **Line Coverage** is employed for *Test Case Generation*. **LLM-As-A-Judge Score** is used to evaluate *Code Understanding*.

## 4. Experiments

### 4.1. Evaluated LLMs and Frameworks

**Benchmarks and Tracks** We evaluate SWE-Compass under Executable and Non-executable tracks. Results are aggregated distribution-aligned over Task Type × Scenario × Language (§3) with fixed seeds.

*Table 2.* Main results by task types on SWE-Compass. AVG is the macro-average across task types. **Abbreviations**: FI=Feature Implementation; FE=Feature Enhancement; BF=Bug Fixing; RF=Refactoring; PO=Performance Optimization; CU=Code Understanding; TG=Test Case Generation; CD=Configuration & Deployment.

| Models | Scores on Different Task Types | | | | | | | | |
|---|---|---|---|---|---|---|---|---|---|
| | FI | FE | BF | RF | PO | CU | TG | CD | AVG |
| **SWE-Agent** | | | | | | | | | |
| Claude-Sonnet-4 | 16.5 | **32.1** | **24.5** | **28.2** | **32.8** | **56.8** | 25.0 | 51.0 | 31.8 |
| Qwen3-Coder-480B-Instruct | 17.9 | 25.4 | 20.3 | 24.1 | 26.4 | 42.1 | 23.4 | 49.5 | 27.2 |
| Kimi-K2-Instruct | 13.7 | 21.0 | 12.7 | 22.4 | 26.4 | 48.2 | 12.4 | 42.5 | 22.7 |
| Gemini-2.5-Pro | 11.8 | 17.4 | 13.1 | 14.9 | 28.9 | 51.8 | 16.8 | 40.5 | 22.4 |
| GPT-4.1 | 14.2 | 16.1 | 13.5 | 19.5 | 21.4 | 43.0 | 18.5 | 39.0 | 21.4 |
| Qwen3-Coder-30B-Instruct | 12.3 | 21.9 | 12.4 | 23.6 | 24.4 | 38.4 | 10.8 | 37.0 | 20.7 |
| Qwen3-235B-A22B-Instruct | 10.4 | 19.2 | 12.2 | 19.5 | 17.9 | 41.1 | 18.5 | 23.0 | 18.8 |
| Gemini-2.5-Flash | 11.3 | 13.4 | 10.1 | 13.2 | 22.4 | 47.7 | 12.6 | 32.0 | 18.5 |
| Deepseek-V3 | 9.4 | 14.3 | 7.7 | 17.2 | 16.9 | 29.2 | 12.8 | 40.0 | 16.5 |
| SWE-agent-LM-32B | 10.4 | 11.6 | 9.0 | 15.5 | 13.9 | 17.9 | 14.7 | 35.0 | 14.7 |
| **Claude Code** | | | | | | | | | |
| Claude-Sonnet-4 | **21.2** | 31.7 | 24.0 | 25.9 | 24.9 | 56.5 | **28.4** | **65.5** | **32.9** |
| Qwen3-Coder-480B-Instruct | 11.8 | 18.3 | 14.4 | 12.1 | 22.9 | 42.8 | 27.3 | 38.5 | 21.9 |
| Qwen3-Coder-30B-Instruct | 17.9 | 23.7 | 12.2 | 21.3 | 15.4 | 36.2 | 21.8 | 28.0 | 21.6 |
| Qwen3-235B-A22B-Instruct | 6.1 | 12.5 | 9.0 | 13.8 | 15.4 | 36.0 | 13.9 | 21.0 | 14.7 |
| Deepseek-V3 | 4.7 | 8.0 | 6.6 | 6.3 | 6.0 | 22.3 | 11.1 | 19.0 | 9.8 |

**Frameworks**  We evaluate two offline workflows: **SWE-Agent** (edit–diff–execute) and **Claude Code** (sandboxed, editor-centric). Both use hardened, network-disabled containers with standardized builds. Workflow details and command matrix are in Appendix A.7.

**Environment, Budgets, and Metrics**  Evaluations use unified budgets in offline containers without retries. We use single-attempt, standard limits, and task-aligned metrics (§3.4). Exact configurations are in Appendix A.7.

**LLMs**  We evaluate **10** models (Appendix A.9): Claude-Sonnet-4, Qwen3-Coder-480B/30B, Qwen3-235B, Kimi-K2, Gemini-2.5-Pro/Flash, GPT-4.1, DeepSeek-V3, and SWE-agent-LM-32B.

### 4.2. Main Results

Table 2 reports Pass@1 by task. Claude-Sonnet-4 ranks first (32.9% Claude Code; 31.8% SWE-Agent). Scores cluster in the low 20s (∼10–33%). Workflows are complementary: of five overlapping models, two favor Claude Code, and three SWE-Agent. Top open-weight model Qwen3-Coder-480B reaches 27.2% (SWE-Agent), trailing proprietary SOTA.

**Findings by different task types.**  A hierarchy emerges (Table 2). Code Understanding (CU) is strong; Configuration & Deployment (CD) shows high variance. Feature Enhancement (FE) and Refactoring (RF) are mid-tier; Feature Implementation (FI) and Bug Fixing (BF) are harder due to localization. Test Case Generation (TG) and Per-

formance Optimization (PO) are challenging. SWE-Agent excels on BF/FI via iterative localization; Claude Code favors deterministic TG/CD; CU is comparable.

**Findings by different frameworks.**  SWE-Agent's edit–diff–execute loop favors investigative, multi-file tasks, despite timeouts. Claude Code excels on well-scoped, deterministic tasks (CD, CU, TG) via lower overhead. Higher scores correlate with more turns (Figure 3, Figure 5), but diminishing returns suggest gains require better localization, not just exploration.

### 4.3. Further Analysis

**Language-level observations.**  Figure 3 and Appendix Table 4 show consistent stratification. JVM/JavaScript score higher; systems languages (C/C++/Rust/Go) are harder; Python is mid-tier (partly due to dataset difficulty). While Claude Code gains on Java/JS for Claude-Sonnet-4, SWE-Agent often matches/outperforms in C# and systems languages. Performance appears governed more by tooling determinism and diagnosability than coding difficulty, highlighting the need for localization/hardening in systems/Python stacks and pruning in JVM/JS. (See Figure 4).

**Interaction turns vs. success.**  Figure 5 shows interaction turn distributions. Deterministic ecosystems (JVM/JS/C#) show lower medians/tighter IQRs under Claude Code, achieving high Pass@1 via reliable signals. Systems languages show heavier tails (especially SWE-Agent) with diminishing returns; Rust is brittle. Python has high vari-

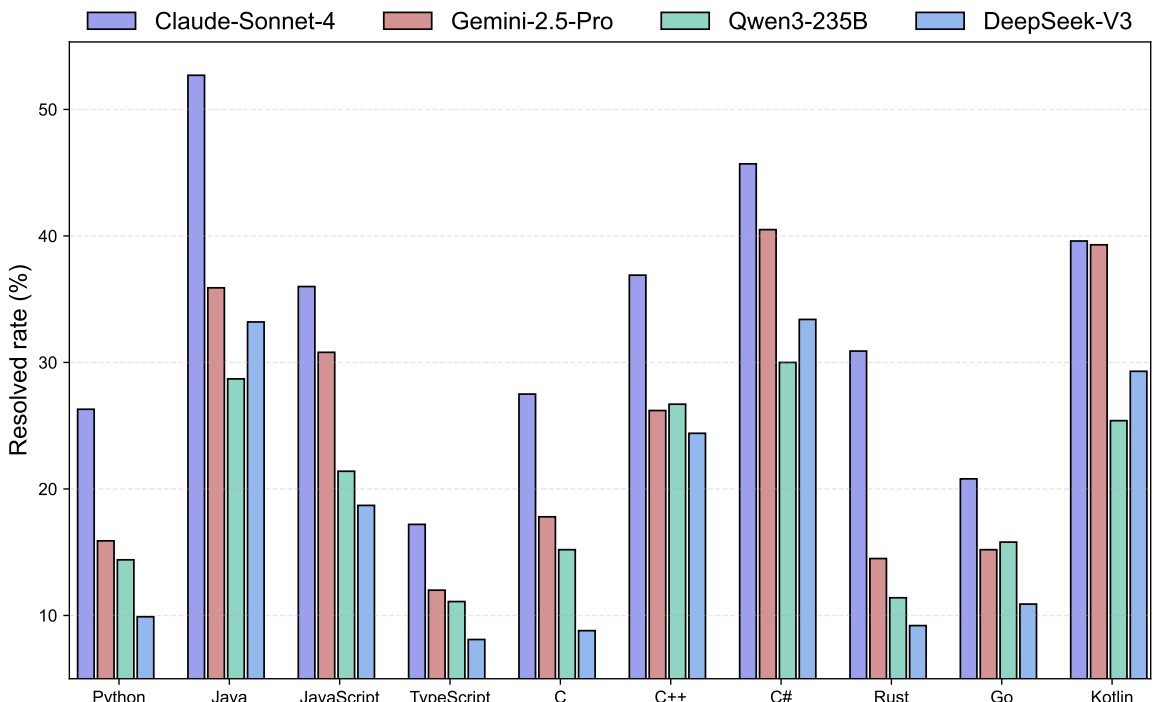

*Figure 3.* Comparison of Pass@1 (%) across the top programming languages for SWE-Agent. Bars represent Pass@1; languages are ordered by overall Pass@1. This plot highlights whether improvements are concentrated in specific languages.

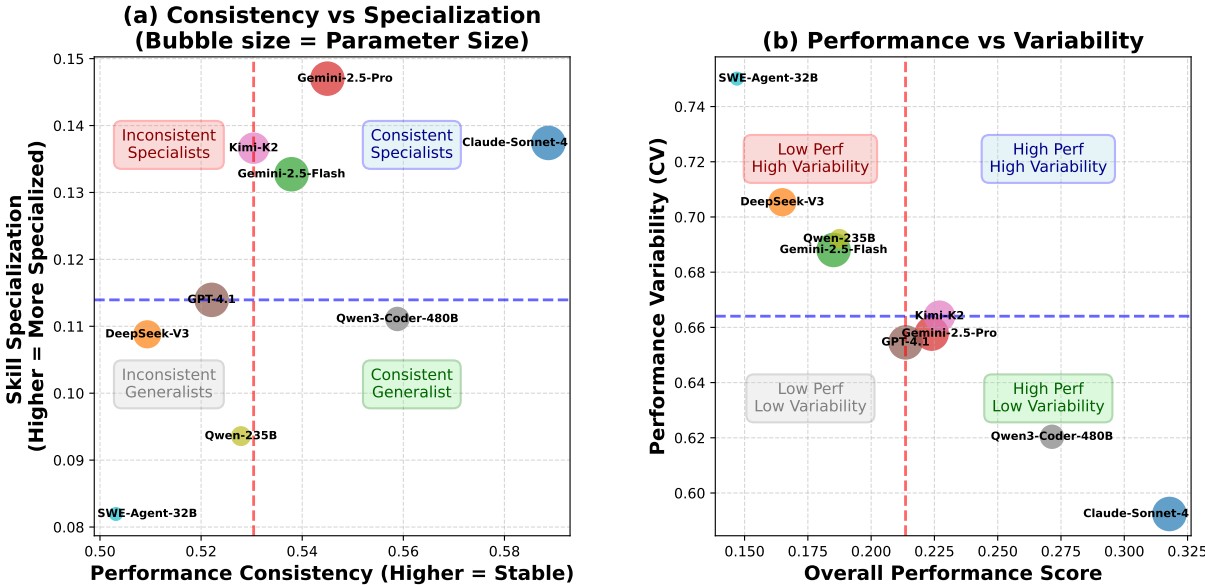

*Figure 4.* (a) Skill specialization vs. performance consistency: Higher values in consistency indicate stability. (b) Performance variability vs. overall performance: Higher overall performance correlates with lower variability (CV).

ance: pinned environments converge quickly (Claude Code), while heterogeneity causes low-yield turns (SWE-Agent). Optimization should prioritize localization for systems languages, hardening for Python, and pruning for JVM/JS.

**Consistency and specialization.** Figure 4 shows that higher-ranked systems improve consistently across languages (panel a), and higher performance correlates with lower variability (panel b). This confirms that top gains stem from broad reliability rather than specialization. Reducing variance—especially in systems languages—is key,

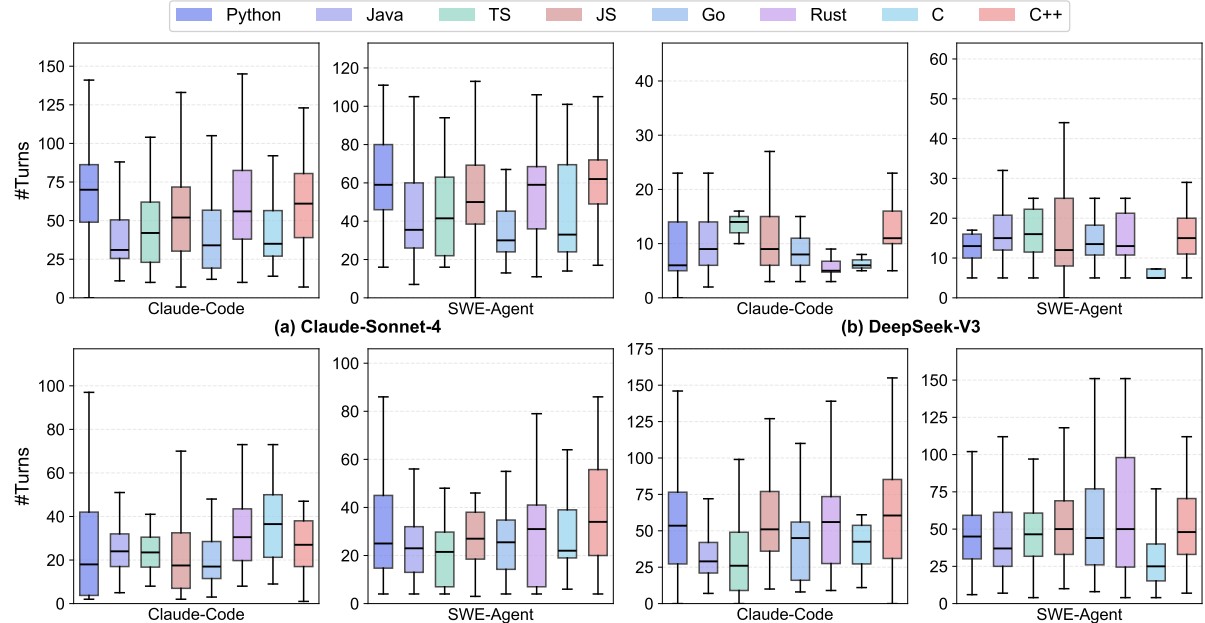

*Figure 5.* Distribution of interaction turns required per language to reveal trade-offs between effort (turns) and success. This highlights whether models achieve high Pass@1 by spending more turns on particular languages.

*Table 3.* Scores on different scenarios: Abbreviations: AD=Application Development; DE=Data Science & Engineering; DS=Database Systems; ID=Infrastructure Development; ML=Machine Learning & AI; SE=Security Engineering; SPD=Specialized Programming Domains; UI/UX=UI/UX Engineering; AVG=macro-average.

| MODEL | Scores on Different Scenarios | | | | | | | | |
|---|---|---|---|---|---|---|---|---|---|
| | AD | DE | DS | ID | ML | SE | SPD | UI/UX | AVG |
| **SWE-Agent** | | | | | | | | | |
| Claude-Sonnet-4 | 31.3 | **33.1** | **28.8** | **29.2** | **35.1** | 31.4 | **29.5** | 38.5 | 31.8 |
| Qwen3-Coder-480B-Instruct | 27.3 | 30.1 | 22.5 | 21.1 | 30.6 | 31.3 | 24.3 | 32.9 | 27.2 |
| Kimi-K2-Instruct | 22.2 | 24.6 | 28.7 | 19.4 | 23.3 | 23.4 | 17.7 | 27.1 | 22.7 |
| Gemini-2.5-Pro | 21.5 | 23.1 | 24.9 | 17.7 | 24.5 | 25.1 | 21.2 | 24.8 | 22.4 |
| GPT-4.1 | 18.8 | 25.5 | 21.9 | 16.3 | 28.5 | 26.5 | 19.4 | 17.8 | 21.4 |
| Qwen3-Coder-30B-Instruct | 20.2 | 25.0 | 21.3 | 17.1 | 24.3 | 23.4 | 16.0 | 21.8 | 20.7 |
| Qwen3-235B-A22B-Instruct | 15.9 | 21.2 | 24.2 | 13.3 | 24.8 | 19.6 | 16.2 | 21.2 | 18.8 |
| Gemini-2.5-Flash | 14.1 | 21.6 | 28.1 | 13.5 | 22.3 | 15.6 | 17.6 | 22.2 | 18.5 |
| Deepseek-V3 | 13.9 | 17.2 | 18.1 | 12.6 | 22.1 | 18.9 | 14.9 | 19.0 | 16.5 |
| SWE-agent-LM-32B | 13.5 | 16.9 | 14.2 | 9.8 | 13.7 | 17.3 | 15.7 | 18.2 | 14.7 |
| **Claude Code** | | | | | | | | | |
| Claude-Sonnet-4 | **33.5** | 32.9 | 27.8 | 26.6 | **33.9** | **37.3** | 29.9 | **44.7** | **32.9** |
| Qwen3-Coder-480B-Instruct | 21.1 | 20.4 | 23.1 | 17.0 | 23.9 | 23.2 | 22.1 | 28.8 | 21.9 |
| Qwen3-Coder-30B-Instruct | 21.2 | 23.3 | 21.6 | 15.3 | 29.2 | 25.3 | 18.3 | 26.8 | 21.6 |
| Qwen3-235B-A22B-Instruct | 12.3 | 17.7 | 19.5 | 14.5 | 14.2 | 19.4 | 8.8 | 13.8 | 14.7 |
| Deepseek-V3 | 7.2 | 14.4 | 10.3 | 9.1 | 11.3 | 11.1 | 7.3 | 9.3 | 9.8 |

and evaluations should report dispersion. Claude-Sonnet-4 exhibits notable consistency.

**Fine-grained scenario analysis.** Table 3 indicates scenario difficulty tracks tooling determinism. High-scoring categories (UI/UX, Security, App Dev) benefit from mature frameworks and fast tests; Claude Code excels here via stable feedback. Complex domains (Database, Infra, ML, Specialized) involve multi-stage builds or non-determinism; SWE-Agent's iterative localization is more resilient but prone to timeouts. Claude Code's advantage concentrates in reliable pipelines. Future systems should: (i) enhance

observability and reproducibility for complex stacks, and (ii) invest in hypothesis pruning for deterministic ones.

Please see the Appendix A.4 for failure mode analysis.

## 5. Conclusion

We introduce SWE-Compass, a unified, execution-grounded benchmark for evaluating agentic coding abilities of large language models. Built from 2,000 real-world GitHub pull requests and spanning diverse tasks, scenarios, and programming languages, SWE-Compass goes beyond bug-fixing–centric evaluations to better reflect realistic software engineering workflows. Empirical results across multiple models and agentic frameworks reveal clear difficulty hierarchies and persistent challenges in requirement grounding and reasoning consistency.

## Acknowledgements

We thank the anonymous reviewers and area chairs for their constructive feedback. This work was supported by the Jiangsu Science and Technology Project under Grant BK20251199 and in part by the National Natural Science Foundation of China under Grant 62506161. We also thank our colleagues and collaborators for helpful discussions and engineering support. Any remaining errors are our own.

## Impact Statement

We have taken several steps to ensure that SWE-Compass adheres to the highest ethical standards in data collection and model evaluation:

**Data Sourcing and Licensing:** All instances in our benchmark are curated from public GitHub repositories. During the data collection phase, we applied strict filtering criteria to ensure that all included repositories carry valid open-source licenses and permit use for research and evaluation purposes.

**Privacy and PII Filtering:** Although the data originates from public PRs and Issues, we conducted a systematic review to ensure that sensitive information, such as Personal Identifiable Information (PII) or private credentials inadvertently left in code patches, was removed or masked during the benchmark construction.

**Environmental Impact:** Recognizing the energy consumption associated with large-scale agentic evaluations, we have standardized build and test commands (e.g., using pinned toolchains and offline caches) to minimize redundant computation and maximize the efficiency of evaluation runs.

**Intended Use and Impact:** The primary goal of SWE-Compass is to provide a rigorous tool for diagnosing and advancing the robustness and safety of AI coding agents. By revealing failure modes like Requirement Misinterpretation and Incomplete Solutions, we hope to guide the development of more reliable and helpful AI assistants for the software engineering community.

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

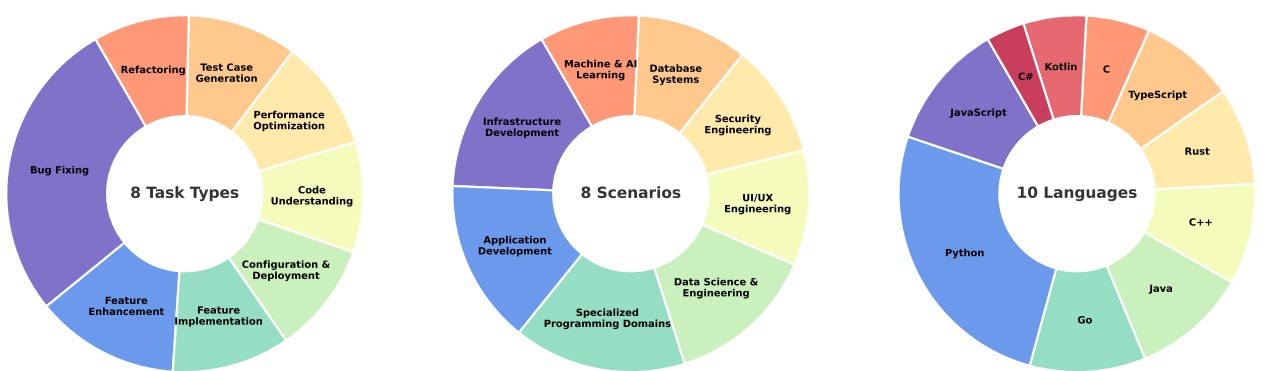

*Figure 6.* Distributions across task types, programming scenarios and languages.

# A. Appendix

### A.1. Limitations

Although SWE-Compass provides a significant advancement in evaluating agentic coding capabilities, several limitations remain, which we have addressed through rigorous experimental design:

**Framework Coverage:** Our current evaluation focuses on two representative agentic frameworks: SWE-Agent and Claude Code. While these do not cover every possible agentic architecture, they represent the state-of-the-art in autonomous coding and provide a robust baseline for comparing model performance across diverse tasks and languages.

**Static Nature of Repositories:** Software ecosystems evolve rapidly. While our 2,000 instances are derived from authentic, high-quality GitHub PRs, they represent a snapshot of developer activity. However, by providing fully reproducible, pinned environments, we ensure that the evaluation remains a stable and rigorous foundation for longitudinal research.

**Computational Cost:** The evaluation of complex agentic frameworks (e.g., SWE-Agent and Claude Code) involves significant multi-turn interactions and containerized builds. This highlights the inherent complexity of real-world software engineering. Rather than a barrier, this limitation underscores the need for future research into agentic efficiency and cost-effective reasoning, which our benchmark is uniquely positioned to measure.

### A.2. Details on Task Types and Programming Scenarios

Task Types:

- *Feature Implementation (FI)*: Developing features or modules from scratch, representing a core activity distinct from modifications or bug fixes.

- *Feature Enhancement (FE)*: Modifying or enhancing existing features to improve functionality, excluding any bug-related changes.

- *Bug Fixing (BF)*: Identifying, diagnosing, and resolving defects in the code, including troubleshooting and debugging.

- *Refactoring (RF)*: Improving the structure and maintainability of the code without altering its external behavior or functionality.

- *Performance Optimization (PO)*: Enhancing system efficiency and resource utilization, focusing specifically on performance improvements and distinct from refactoring.

- *Code Understanding (CU)*: Exploring, analyzing, and understanding code through static and dynamic analysis, including generating reports.

- *Test Case Generation (TG)*: Automatically generating unit and integration tests to validate code and ensure quality assurance.

- *Configuration & Deployment (CD)*: Setting up environments, managing dependencies, and writing deployment scripts to ensure smooth application operation.

Programming Scenarios:

- *Application Development (AD)*: Developing applications for specific environments such as web or desktop platforms, with an emphasis on feature implementation and platform adaptation.

- *Database Systems (DS)*: Designing, developing, managing, and optimizing databases to ensure efficient data storage, access, and consistency.

- *Data Science & Engineering (DE)*: Handling data processing, analysis, mining, ETL, and feature engineering, emphasizing data-driven decision-making and efficient pipeline construction.

- *Machine Learning & AI (ML)*: Training models, building recommendation systems, applying algorithms to enable intelligent decision-making and predictions.

- *Infrastructure Development (ID)*: Building foundational systems such as distributed architectures, system deployment, and DevOps tools, emphasizing stability, scalability, and automation.

- *Specialized Programming Domains (SPD)*: Addressing areas such as graphics, gaming, multimedia, and networking that require specialized technical expertise and tailored solutions.

- *Security Engineering (SE)*: Ensuring application and system security, identifying vulnerabilities, and implementing measures such as encryption to maintain compliance with security standards.

- *UI/UX Engineering (UI/UX)*: Designing and optimizing user interfaces and experiences across platforms to enhance visual appeal, usability, and consistency.

Programming Languages: Python, JavaScript, TypeScript, Java, C, C++, Go, Rust, Kotlin, C#.

## A.3. Evaluation Metrics

For each type of task, we select appropriate evaluation metrics to measure the model's performance. These include:

1. **Pass@1**: The fraction of resolved samples achieved under a single attempt with fixed decoding and resource budgets.

2. **Performance Optimization Score**: A binary indicator (0/1). The score is 1 if the model's optimized code passes a single test and the time spent on execution is less than 80% of the time taken by the unoptimized code; otherwise, the score is 0.

3. **Line Coverage**: This metric evaluates the extent to which the program code has been executed during test case execution. The formula for calculating line coverage is:

$$\frac{\text{Number of Executed Code Lines}}{\text{Total Number of Code Lines}} \times 100\%$$

4. **LLM-As-A-Judge Score**: Following (Zheng et al., 2023a; Zhang et al., 2025a;b; Li et al., 2025)), we use a large language model (LLM) to review the model output according to a checklist; the final score is the proportion of checkpoints passed by the model output.

For specific tasks, the following metrics are used. For **Feature Implementation, Feature Enhancement, Bug Fixing, and Refactoring**, **Pass@1** is used to measure the model's performance. For **Performance Optimization**, the **Performance Optimization Score** is used to evaluate the model's performance. For **Test Case Generation**, we employ **Line Coverage** to assess the quality of the test cases generated by the model. In our implementation, we use pytest (Pajankar, 2017) to compute line coverage for Python. For TypeScript and JavaScript, we use C8 (Vassudanagunta, 2025) to calculate line coverage. For **Code Understanding**, we use the **LLM-As-A-Judge Score** to evaluate the accuracy of the model's understanding of the code.

**Trajectory Failure Modes Analysis**

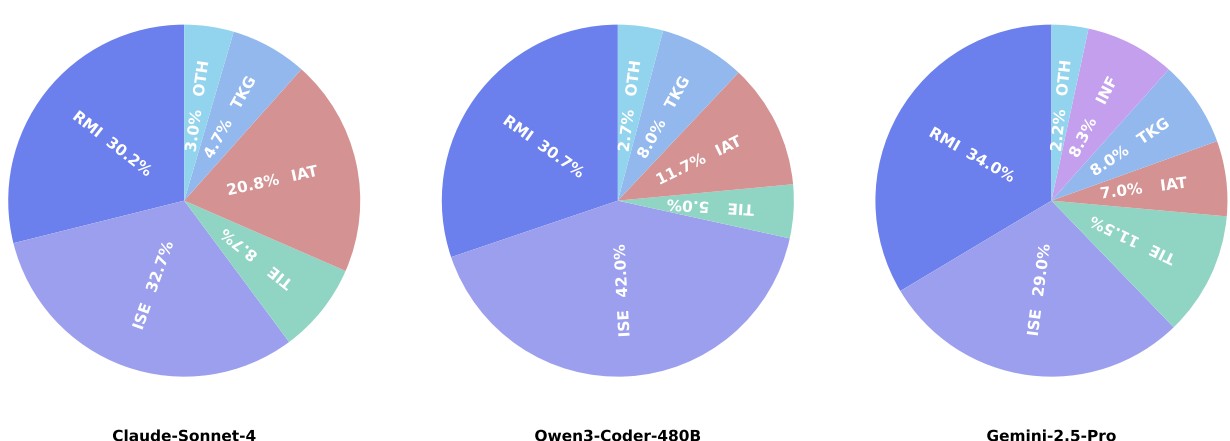

*Figure 7.* Distribution of trajectory failure modes on SWE-Compass. Abbreviations: RMI=Requirement Misinterpretation, ISE=Incomplete Solution & Side Effects, TIE=Tool Invocation Error, IAT=Inadequate Testing, TKG=Technical Knowledge Gap, INF=Infinite Loop, OTH=Others.

---

**Judge Prompt for Code Understanding Task**

```
Evaluate if the answer satisfies the question requirements using a natural language
    explanation.

QUESTION:
{question_text}

REQUIREMENTS (checklist items):
{chr(10).join(checklist_text)}
{patch_section}
ANSWER:
{truncated_answer}

EVALUATION RULES:
1. Answer MUST use clear English explanations, NOT just code diffs, or other types
    of content
2. Only give 1.0 when ALL checklist items are thoroughly satisfied with clear
    explanations.
3. Score = (satisfied items) / (total items)
4. Penalize: code diffs without explanation, vague statements, wrong info. Give 0.0
    when the answer is just code diffs or completely wrong.

JSON response:
{{
  "reasoning": "Brief explanation of which items satisfied/unsatisfied and why",
  "score": ,
  "satisfied_items": ["item_id1", ...]
}}
```

### A.4. Failure Mode Analysis

As shown in Figure 7, to systematically understand the limitations of current coding agents, we perform a post-hoc failure analysis on **SWE-Agent** trajectories from our SWE-Compass benchmark. Following (Yang et al., 2024)—who report 87% agreement between automated LLM judges and human experts—we adopt an *LLM-as-Judge* protocol with **Claude-Sonnet-4** as the judge; the exact prompt is provided in Appendix A.6. We sample **600** failed trajectories *per model* for three

representative systems: **Claude-Sonnet-4**, **Qwen3-Coder-480B**, and **Gemini-2.5-Pro**.

Specifically, through manual inspection of submitted-but-failed trajectories, we develop a comprehensive six-category taxonomy capturing actual root causes:

1. **Requirement Misinterpretation**: The agent failed to properly understand and locate the problem, including misidentifying affected files, misjudging severity, confusing problem types, or failing to identify root causes and understand dependencies, data flow, or system architecture.

2. **Inadequate Testing**: The agent provided incomplete test coverage, missing edge cases, compatibility issues, performance impacts, integration scenarios, or multi-platform testing requirements.

3. **Incomplete Solution & Side Effects**: The agent provided an incomplete fix that only addressed symptoms rather than root causes, or introduced new issues, including regressions, security vulnerabilities, environment configuration errors, data corruption risks, or breaking changes to existing functionality.

4. **Technical Knowledge Gap**: The agent demonstrated insufficient technical proficiency or violated domain-specific conventions, including lacking necessary knowledge in specialized domains (UI/frontend, security, accessibility, DevOps, performance, analytics) or incorrectly handling domain-specific issues (data processing, security implementations, UI/UX standards, API design, documentation synchronization).

5. **Tool Invocation Error**: The agent encountered errors while using tools due to incorrect syntax, context overflow from file operations, or parse/analysis tool failures.

6. **Infinite Loop**: The agent got stuck in loops without convergence, including repeated attempts at the same solution, oscillating between decisions, or endlessly reading files without making progress.

Note that **OTH (Other)** denotes rare cases, which are not covered by the above taxonomy (e.g., corrupted artifacts or external executor glitches). The figure caption lists all abbreviations for completeness. As shown in Figure 7, we report the per-model distribution of failure modes on SWE-Compass. Based on **600** error traces per model, we draw the following conclusions: (1) **Shared bottlenecks in comprehension and implementation**. All models exhibit high error rates in **Requirement Misinterpretation** (30–34%) and **Incomplete Solution & Side Effects** (29–42%), together accounting for $> 60\%$ of failures. By contrast, **Technical Knowledge Gap** is consistently low (5–8%), suggesting the core limitations lie in requirement grounding and holistic solution design rather than basic coding proficiency. (2) **Distinct model characteristics**. *Claude-Sonnet-4* is the most balanced, showing the lowest **Technical Knowledge Gap** (4.7%) but room to improve on **Inadequate Testing** (20.8%). *Qwen3-Coder-480B* has the highest **Incomplete Solution & Side Effects** rate (42%, vs. Claude-Sonnet-4's 32.7%), revealing weaknesses in end-to-end design. *Gemini-2.5-Pro* shows the highest **Requirement Misinterpretation** (34%) and a notable **Infinite Loop** issue (8.3%), posing reliability risks in production.

### A.5. Claude Code: Parallel Tool-Calls System Prompt

The following prompt is appended via SDK to encourage parallel tool invocations when operations are independent.

---

**Claude Code: Parallel Tool-Calls System Prompt**

```
PARALLEL_TOOL_CALLS_SYSTEM_PROMPT = """
<use_parallel_tool_calls>
For maximum efficiency, whenever you perform multiple independent operations, invoke
    all relevant tools simultaneously rather than sequentially. Prioritize calling
    tools in parallel whenever possible. For example, when reading 3 files, run 3
    tool calls in parallel to read all 3 files into context at the same time. When
    running multiple read-only commands like `ls` or `list_dir`, always run all of
    the commands in parallel. Err on the side of maximizing parallel tool calls
    rather than running too many tools sequentially.
</use_parallel_tool_calls>
"""

options = ClaudeCodeOptions(
```

---

```
  max_turns=150,
  cwd=Path(working_dir),
  permission_mode="bypassPermissions",
  append_system_prompt=PARALLEL_TOOL_CALLS_SYSTEM_PROMPT.strip()
)
```

## A.6. Trajectory Failure Analysis Prompt

**Trajectory Failure Analysis Prompt**

### System Prompt:

You are an expert software engineer analyzing why a software engineering
agent failed to resolve an issue.

AVAILABLE AGENT ACTIONS:

---- BEGIN FUNCTION #1: bash ----
Description: Execute a bash command in the terminal.
* Can generate very large outputs when listing files (ls, find, grep)
* Output contributes directly to context window usage
* Commands like 'find /repo -name "*.py"' can list thousands of files
* Large outputs can quickly fill the context window

Parameters:
  (1) command (string, required): The bash command to execute. Can be empty to view
      additional logs when previous exit code is `-1`. Can be `ctrl+c` to interrupt
      the currently running process.
---- END FUNCTION #1 ----

---- BEGIN FUNCTION #2: submit ----
Description: Finish the interaction when the task is complete OR if the assistant
    cannot proceed further with the task.
* Used when agent thinks task is done (may be correct or incorrect solution)
* Also used when agent is stuck and cannot make progress
* No parameters are required for this function.
---- END FUNCTION #2 ----

---- BEGIN FUNCTION #3: str_replace_editor ----
Description: Custom editing tool for viewing, creating and editing files
* State is persistent across command calls and discussions with the user
* If `path` is a file, `view` displays the result of applying `cat -n`. If `path` is
      a directory, `view` lists non-hidden files and directories up to 2 levels deep
* Directory views can generate large outputs contributing to context usage
* The `create` command cannot be used if the specified `path` already exists as a
    file
* If a `command` generates a long output, it will be truncated and marked with `<
    response clipped>`
* The `undo_edit` command will revert the last edit made to the file at `path`

Notes for using the `str_replace` command:
* The `old_str` parameter should match EXACTLY one or more consecutive lines from
    the original file. Be mindful of whitespaces!
* If the `old_str` parameter is not unique in the file, the replacement will not be
    performed. Make sure to include enough context in `old_str` to make it unique
* The `new_str` parameter should contain the edited lines that should
  replace the `old_str`

Parameters:
  (1) command (string, required): The commands to run. Allowed options are:
      `view`, `create`, `str_replace`, `insert`, `undo_edit`.

```
    (2) path (string, required): Absolute path to file or directory,
        e.g. '/repo/file.py' or '/repo'.
    (3) file_text (string, optional): Required parameter of 'create' command,
        with the content of the file to be created.
    (4) old_str (string, optional): Required parameter of 'str_replace' command
        containing the string in 'path' to replace.
    (5) new_str (string, optional): Optional parameter of 'str_replace' command
        containing the new string (if not given, no string will be added).
        Required parameter of 'insert' command containing the string to insert.
    (6) insert_line (integer, optional): Required parameter of 'insert' command.
        The 'new_str' will be inserted AFTER the line 'insert_line' of 'path'.
    (7) view_range (array, optional): Optional parameter of 'view' command when
        'path' points to a file. If none is given, the full file is shown.
        If provided, the file will be shown in the indicated line number range,
        e.g. [11, 12] will show lines 11 and 12. Indexing at 1 to start.
        Setting '[start_line, -1]' shows all lines from 'start_line' to the
        end of the file.
---- END FUNCTION #3 ----

---- BEGIN FUNCTION #4: file_viewer ----
Description: Interactive file viewer for opening and navigating files in
the editor.
* open <path> [<line_number>]: Opens the file at path. If line_number is
  provided, the view moves to include that line.
* goto <line_number>: Moves the window to show the specified line number.
* scroll_down: Moves the window down 100 lines.
* scroll_up: Moves the window up 100 lines.

Parameters:
    (1) command (string, required): One of 'open', 'goto', 'scroll_down',
        'scroll_up'.
    (2) path_or_line (string/int, optional): For 'open', a path (and optional
        line). For 'goto', a line number.
---- END FUNCTION #4 ----

---- BEGIN FUNCTION #5: search_tools ----
Description: Searching utilities for locating text or files within the
workspace.
* search_file <search_term> [<file>]: Searches for search_term in file.
  If file is not provided, searches the current open file.
* search_dir <search_term> [<dir>]: Searches for search_term in all files
  in dir. If dir is not provided, searches in the current directory.
* find_file <file_name> [<dir>]: Finds all files with the given name in dir.
  If dir is not provided, searches in the current directory.

Parameters:
    (1) subcommand (string, required): One of 'search_file', 'search_dir',
        'find_file'.
    (2) arg1 (string, required): The search term or file name, depending on
        subcommand.
    (3) arg2 (string, optional): Target file (for search_file) or directory
        (for search_dir/find_file).
---- END FUNCTION #5 ----

---- BEGIN FUNCTION #6: edit_block ----
Description: Block editor for replacing ranges in the current open file
and finalizing edits.
* edit <n>:<m> <replacement_text>: Replaces lines n through m (inclusive)
  with the given text in the open file. Ensure indentation is correct.
* end_of_edit: Applies the pending changes. Python files are syntax-checked
  after the edit; if an error is found, the edit is rejected.

Parameters:
```

```
  (1) command (string, required): 'edit' or 'end_of_edit'.
  (2) range_and_text (varies): For 'edit', a line range 'n:m' and the
      replacement text.
---- END FUNCTION #6 ----

---- BEGIN FUNCTION #7: create_file ----
Description: Creates and opens a new file with the given name.

Parameters:
  (1) filename (string, required): Absolute or workspace-relative path to
      create. The file must not already exist.
---- END FUNCTION #7 ----

##PROBLEM STATEMENT##
{problem_statement}

##TRAJECTORY SUMMARY##
- Total steps: {total_steps}
- Final state: Failed (no successful patch generated / failed on some unit test)

##ANALYSIS INSTRUCTIONS##

**IMPORTANT: This trajectory FAILED in final evaluation. The agent likely
believed it succeeded, but it was WRONG.**

The agent may have:
- Claimed the issue was resolved or fixed
- Written custom tests that passed
- Expressed high confidence in the solution
- Stated "the implementation is complete" or "all tests pass"
- Created demo scripts showing the fix "works"
- Manually verified outputs that looked correct

Despite these apparent indicators of success, the final evaluation proves
the solution was INCORRECT. Therefore, ignore the agent's self-assessment
and focus on identifying the actual flaws. Select ONE category below that
best describes the actual flaw:
Requirement Misinterpretation: The agent failed to properly understand and
locate the problem, including misidentifying affected files, misjudging
severity, confusing problem types, or failing to identify root causes and
understand dependencies, data flow, or system architecture.
Inadequate Testing: The agent provided incomplete test coverage, missing
edge cases, compatibility issues, performance impacts, integration scenarios,
or multi-platform testing requirements.
Incomplete Solution & Side Effects: The agent provided an incomplete fix
that only addressed symptoms rather than root causes, or introduced new
issues including regressions, security vulnerabilities, environment
configuration errors, data corruption risks, or breaking changes to existing
functionality.
Technical Knowledge Gap: The agent demonstrated insufficient technical
proficiency or violated domain-specific conventions, including lacking
necessary knowledge in specialized domains (UI/frontend, security,
accessibility, DevOps, i18n, performance, analytics) or incorrectly handling
domain-specific issues (data processing, security implementations, UI/UX
standards, API design, documentation synchronization).
Tool Invocation Error: The agent encountered errors while using tools due
to incorrect syntax, context overflow from file operations, or parse/analysis
tool failures.
infinite_loop: The agent got stuck in loops without convergence, including
repeated attempts at the same solution, oscillating between decisions, or
endlessly reading files without making progress.

other: The agent failed to resolve the issue for reasons not covered by the
```

```
above categories.

Do NOT invent or propose new categories. If none fits, use "other". Category
must be all lowercase with underscores. Remember to write two new lines
before the category.
\end{verbatim}

\textbf{User Prompt:}

\begin{verbatim}
##INSTANCE INFORMATION##
Instance ID: {instance_id}

##The complete trajectory of the interaction (to be analyzed)##
{traj_text}

##OUTPUT FORMAT##
You MUST provide your response in this exact format:
<description>
xxx
</description>

<category>
xxx
</category>

<error_actions>
If the Assistant gets stuck in a loop or encounters a tool_error error, indicate the
    incorrect action and parameters. If the Assistant misunderstands the question,
    set error_action="None".
</error_actions>
```

## A.7. Executor Details and Method-Specific Settings

Unless otherwise noted, all runs are strictly offline. Below we record method-specific configurations referenced in §4:

- **SWE-Agent.** *max turns* $= 150$; per-tool step timeout $= 600$ s; `parse_function` set to *function calling*; long observations truncated; compiled artifacts filtered via ".gitignore"; language-specific build/test commands repaired for stability.

- **Claude Code.** *max turns* $= 150$; `permission_mode` $=$ **bypassPermissions**; system prompt appended to encourage parallel tool calls (Appx. A.5); may internally invoke a SubAgent; networking disabled.

**Standardized offline build/test commands (per language).** We standardize non-interactive commands to ensure reproducible builds and comparable feedback signals across languages:

- **Python**: `pytest -q`

- **JavaScript/TypeScript**: `npm ci && npm test -run`

- **Java**: `mvn -B -DskipTests=false test`

- **Go**: `go test ./...`

- **Rust**: `cargo test -locked`

- **C/C++**: `cmake -build && ctest -j1`

*Table 4.* Top-10 languages: Pass@1 (%) per model. Columns are languages; rows are models grouped by agent.

| MODEL | Scores on Different Language | | | | | | | | | | |
|---|---|---|---|---|---|---|---|---|---|---|---|
| | **Python** | **Java** | **JavaScript** | **TypeScript** | **C** | **C++** | **C#** | **Rust** | **Go** | **Kotlin** | **AVG** |
| *SWE-Agent* | | | | | | | | | | | |
| Claude-Sonnet-4 | **26.3** | 52.7 | 36.0 | 17.2 | **27.5** | **36.9** | 45.7 | **30.9** | 20.8 | 39.6 | 31.8 |
| Qwen3-Coder-480B-Instruct | 20.8 | 50.0 | 36.0 | 13.2 | 18.0 | 30.6 | 30.9 | 24.6 | 16.2 | 38.1 | 27.2 |
| Kimi-K2-Instruct | 13.2 | 43.0 | 24.4 | 13.7 | 15.8 | 33.4 | 42.2 | 15.2 | **21.9** | 28.2 | 22.7 |
| Gemini-2.5-Pro | 15.9 | 35.9 | 30.8 | 12.0 | 17.8 | 26.2 | 40.5 | 14.5 | 15.2 | 39.3 | 22.4 |
| GPT-4.1 | 17.2 | 31.5 | 25.0 | 10.0 | 16.2 | 28.3 | **46.8** | 15.8 | 12.7 | 34.7 | 21.4 |
| Qwen3-Coder-30B-Instruct | 13.5 | 32.1 | 26.3 | 13.7 | 13.6 | 28.7 | 28.4 | 14.9 | 15.5 | 37.8 | 20.7 |
| Qwen3-235B-A22B-Instruct | 14.4 | 28.7 | 21.4 | 11.1 | 15.2 | 26.7 | 30.0 | 11.4 | 15.8 | 25.4 | 18.8 |
| Gemini-2.5-Flash | 12.7 | 30.4 | 20.5 | 14.1 | 15.7 | 24.9 | 32.2 | 13.1 | 16.0 | 22.3 | 18.5 |
| Deepseek-V3 | 9.9 | 33.2 | 18.7 | 8.1 | 8.8 | 24.4 | 33.4 | 9.2 | 10.9 | 29.3 | 16.5 |
| SWE-agent-LM-32B | 11.1 | 30.5 | 20.8 | 7.2 | 4.7 | 17.0 | 20.9 | 6.5 | 11.7 | 18.7 | 14.7 |
| *Claude Code* | | | | | | | | | | | |
| Claude-Sonnet-4 | 23.5 | **60.0** | **46.3** | **18.5** | 24.5 | 34.5 | 43.2 | 28.2 | 18.2 | **51.9** | **32.9** |
| Qwen3-Coder-480B-Instruct | 17.7 | 39.4 | 30.3 | 15.2 | 14.9 | 21.1 | 27.4 | 20.7 | 13.2 | 21.4 | 21.9 |
| Qwen3-Coder-30B-Instruct | 16.0 | 41.1 | 28.8 | 13.6 | 16.2 | 22.1 | 23.7 | 11.5 | 20.7 | 33.3 | 21.6 |
| Qwen3-235B-A22B-Instruct | 10.4 | 18.7 | 21.5 | 9.0 | 12.3 | 14.5 | 20.3 | 11.8 | 12.4 | 32.1 | 14.7 |
| Deepseek-V3 | 7.1 | 14.4 | 15.0 | 4.1 | 5.7 | 11.7 | 16.0 | 4.7 | 8.0 | 20.6 | 9.8 |

**Execution hardening and navigation controls.** To minimize offline flakiness and improve determinism, we apply:

- Pinned toolchains inside containers; pre-populated offline caches/proxies for `pip`, `npm`, `cargo`, Maven/Gradle, and Go modules.

- Truncation of long observations and logs; filtering of binaries and dev servers via ".gitignore".

- Normalization of EOL/encoding (LF, UTF-8); `git safe.directory` set; whitespace-tolerant patching.

- Repository navigation pruning by extensions; optional function/class extraction (read-only) to speed up localization.

- Budgets and quotas: max 150 turns; per-step 600 s; global job limits; auto-kill of long-lived processes.

- Language-specific repairs for brittle stacks (e.g., Java multi-module builds, Node lockfile drift, Rust workspaces, C/C++ out-of-tree builds).

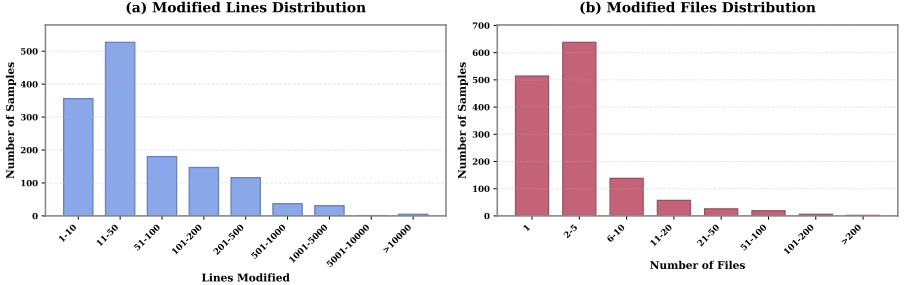

*Figure 8.* Distribution of modified files and lines in golden patches, aggregated by benchmark slices and task types. The breakdown shows the scope and granularity of code changes across different programming tasks.

## A.8. Analysis of Open-Source Benchmark Distributions

We annotated several repository-level SWE benchmark datasets, including **SWE-bench-Verified** ($n = 500$), **SWE-bench-Live** ($n = 500$), **SWE-bench-Multilingual** ($n = 300$), **SWE-bench-Pro** ($n = 731$), and **SWE-rebench** ($n = 449$),

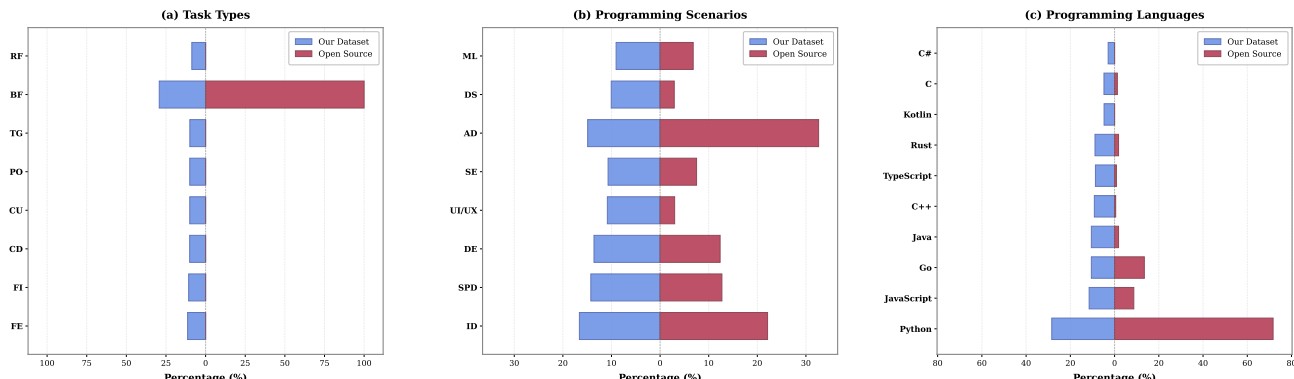

*Figure 9.* Distributions across task types, programming scenarios, and languages in Open-Source SWE Benchmarks and Github PR & Issue. **Abbreviations:** FE: Feature Enhancement, FI: Feature Implementation, CD: Configuration & Deployment, CU: Code Understanding, PO: Performance Optimization, TG: Test Case Generation, BF: Bug Fixing, RF: Refactoring; ID: Infrastructure Development, SPD: Specialized Programming Domains, DE: Data Science & Engineering, SE: Security Engineering, AD: Application Development, DS: Database Systems, ML: Machine Learning & AI, UI/UX: UI/UX Engineering.

totaling **2,480** instances across multiple programming languages and scenarios. Figure 9(a) presents the distribution of these open-source datasets across task types, program scenarios, and languages. Through detailed analysis, we identified the following limitations:

- **Incomplete Task Type Coverage.** Existing benchmarks are entirely focused on *Bug Fixing* tasks, which comprise 100% of all instances. In contrast, several important task types—including *Feature Enhancement*, *Feature Implementation*, *Configuration & Deployment*, *Code Understanding*, *Performance Optimization*, *Test Case Generation*, and *Refactoring*—are completely absent.

- **Imbalanced Scenario Distribution.** A large portion of the data focuses on *Application Development* (32.6%), whereas other critical scenarios such as *UI/UX Engineering* (3.0%), *Database Systems* (2.9%), and *Security Engineering* (7.5%) receive significantly less coverage. Meanwhile, *Infrastructure Development* accounts for 22.1%.

- **Severe Programming Language Imbalance.** The datasets are overwhelmingly dominated by *Python* (71.7%), with minimal coverage of other programming languages such as *Go* (13.5%), *JavaScript* (8.7%), and others combined accounting for less than 6%.

### A.9. Model List

We evaluate our approach using a diverse set of state-of-the-art language models, including both closed-source and open-source models. The closed-source models include Claude-Sonnet-4-20250514(Anthropic, 2025), Gemini-2.5-Flash, Gemini-2.5-Pro(Gemini Team & Google, 2023), and GPT-4.1-2025-04-14(OpenAI et al., 2024). For open-source models, we utilize Qwen3-Coder series(Team, 2025; Hui et al., 2024a), Kimi-K2-Instruct-0905(Team et al., 2025b), Deepseek-V3-0324(Liu et al., 2024a), and SWE-agent-LM-32B(Yang et al., 2025c). The complete list of models with their official links is provided in Table 5.

### A.10. Future Works

We see several directions to extend SWE-Compass and strengthen the community's ability to measure and drive progress:

- **Scale and coverage**. Expand the dataset size, languages (e.g., mobile stacks and diverse SQL dialects), and repository types (monorepos, polyglot services), while maintaining distribution alignment across task, scenario, language, and difficulty.

- **Harder long-context settings**. Introduce multi-module, cross-process, and build-pipeline tasks that stress architectural coherence, multi-file reasoning, and cross-session memory under strict executability.

*Table 5.* Model List

| Model | Link |
|---|---|
| *Closed-Source Models* | |
| Claude-Sonnet-4-20250514 | https://www.anthropic.com/claude |
| Gemini-2.5-Flash | https://deepmind.google/technologies/gemini/flash/ |
| Gemini-2.5-Pro | https://deepmind.google/technologies/gemini/pro/ |
| GPT-4.1-2025-04-14 | https://openai.com/gpt-4 |
| *Open-Source Models* | |
| Qwen3-Coder-480B-A35B-Instruct | https://huggingface.co/Qwen/Qwen3-Coder-480B-A35B-Instruct |
| Qwen3-Coder-30B-A3B-Instruct | https://huggingface.co/Qwen/Qwen3-Coder-30B-A3B-Instruct |
| Qwen3-235B-A22B-Instruct-2507 | https://huggingface.co/Qwen/Qwen3-235B-A22B-Instruct-2507 |
| Kimi-K2-Instruct-0905 | https://huggingface.co/moonshotai/Kimi-K2-Instruct-0905 |
| Deepseek-V3-0324 | https://huggingface.co/deepseek-ai/DeepSeek-V3 |
| SWE-agent-LM-32B | https://huggingface.co/SWE-bench/SWE-agent-LM-32B |

- **Metrics and protocols**. Enrich task-type–aligned metrics with long-context diagnostics (e.g., consistency and variance reporting), stabilize timing/coverage signals, and unify cost/efficiency reporting (turns, wall-clock, tool invocations) under fixed budgets.

- **Evaluation tracks**. Explore safe online or incremental tracks with evolving repositories and dependency drift, paired with sandboxing, artifact isolation, and replayable logs for fair comparison over time.

- **Human-in-the-loop calibration**. Establish human adjudication subsets and reliability audits for LLM-as-Judge, improving rubric calibration and model–human agreement on non-executable tasks.

- **Reproducibility, safety, and accessibility**. Continue releasing containers, minimal repro scripts, and a lightweight subset with stable seeds; strengthen privacy/safety filtering and provide clearer documentation to lower the barrier to participation.

