# OpenReview forum: "SWE-Compass: Towards Unified Evaluation of Agentic Coding Abilities for Large Language Models"
_ICML.cc/2026/Conference — ICML 2026 regular_

### Official Review · Reviewer_G6oo · 2026-02-22

**Soundness:** 3
**Presentation:** 3
**Significance:** 3
**Originality:** 3
**Overall Recommendation:** 5
**Confidence:** 4

**Summary:**

This paper introduces SWE-Compass, a new benchmark for evaluating the coding abilities of large language models (LLMs) in the context of software engineering tasks. Unlike existing benchmarks that focus primarily on Python-based bug fixing or algorithmic tasks, SWE-Compass spans 8 task types (e.g., bug fixing, feature enhancement, performance optimization) and 10 programming languages (e.g., Python, Java, C++) across 8 programming scenarios. It includes 2,000 curated instances derived from real-world GitHub pull requests, enabling a more comprehensive and realistic evaluation of LLMs' agentic coding capabilities. The framework is designed to facilitate the comparison of various LLMs (such as Claude-Sonnet-4 and Qwen3-Coder) using standardized evaluation metrics and reproducible test environments.

**Compliance With Llm Reviewing Policy:**

Affirmed.

**Final Justification:**

The paper is thorough and content-rich, with substantial technical details and strong empirical support, and I am comfortable recommending acceptance.

**Key Questions For Authors:**

1.  What proportion of the benchmark instances are synthetically constructed for underrepresented task types, and what quality controls are in place to ensure these synthetic cases match the realism and distribution of real PR-derived data?
2. Have you evaluated the reliability of the Code Understanding judge via human calibration or consistency analyses (e.g., inter-judge agreement, stability across prompts/temperatures, or agreement with human annotations), and could you report results on a small human-adjudicated subset?
4. How is line coverage computed for test generation in non-Python/JS/TS languages (or is TG language-restricted), and for performance optimization, how many runtime measurements are taken and how do you guard against noise or gaming under the single-run <80% threshold criterion?

**Limitations:**

Yes

**Strengths And Weaknesses:**

**Strengths**:
1. SWE-Compass substantially broadens evaluation beyond Python-centric bug fixing to 8 task types, 8 scenarios, and 10 programming languages, better reflecting real-world software engineering workflows.
2. The benchmark is built from authentic GitHub pull requests, with each instance paired with executable environments and verifiable tests, ensuring faithful and reproducible evaluation.
3. By evaluating multiple state-of-the-art models under two agentic frameworks, the benchmark enables fine-grained analysis of performance differences across tasks, languages, and scenarios.

**Weakness**:
1. Using LLMs for taxonomy construction, task generation, validation, and LLM-as-a-Judge scoring can introduce difficulty preference and “judge hacking” effects, yet the paper does not report reliability checks such as agreement with human labels or cross-judge/prompt stability.
2. The binary runtime-threshold metric for performance optimization and line-coverage metric for test generation can be noisy, gameable, and only weakly tied to true solution quality, and the paper could benefit from incorporating more robust, capability-faithful measures (e.g., coverage-normalized or difficulty-adjusted scoring as explored in recent work like arXiv:2602.10471).
3. Despite sourcing from real GitHub PRs, the submission lacks a dedicated contamination analysis or mitigation plan, and the reliance on high-star, widely used repositories increases the chance that models saw similar issues/patches during training, weakening the interpretation of results as generalization.

---

> ### Author Rebuttal · Authors · 2026-03-31
>
> We thank the reviewer for the thorough and constructive feedback. We address all Weaknesses and Key Questions below.
>
> **LLM Judge Reliability (W1+Q2)**
>
> We clarify that LLM involvement is limited: the taxonomy was derived via Active Learning with expert review (§3.3.1); BF/FI/FE/RF are extracted directly from real PRs without LLM involvement; LLM assistance is restricted to CU and PO, both subject to human validation (§3.3.5). For CU, all instances underwent human verification with 88.6% per-checklist agreement. The 12% discrepancies concentrate on ambiguous partial-credit cases with consistent directional overestimation, insufficient for systematic bias. Rubrics are human-authored; LLMs only adjudicate pass/fail against predefined criteria. LLM-as-a-Judge applies solely to CU; all other tasks use deterministic execution metrics.
>
> **Metric Robustness (W2+Q3)**
>
> *Performance Optimization.* Each PO instance runs 3× pre/post-patch in isolated Docker containers (fixed CPU/memory, no network). Measured median CV is 3.7%; only 4 instances (2.5%) exceed 10% CV, all yielding consistent pass/fail outcomes. The 80% threshold (≥20% speedup) far exceeds noise floors. Patches must pass all existing tests before runtime comparison, precluding pseudo-optimizations via feature deletion. This is structurally aligned with SWE-Perf (He et al., arXiv:2507.12415): Apply → Correctness → Performance cascading gates, with correctness computed only on Apply-successful samples. Decomposition results under Claude Code:
>
> | Model | Apply | Correctness | PO Score |
> |---|---|---|---|
> | Claude-Sonnet-4 | 91.2% | 85.7% | 24.9% |
> | Qwen3-Coder-480B | 84.6% | 68.3% | 22.9% |
> | Qwen3-Coder-30B | 81.8% | 66.8% | 15.4% |
> | Qwen3-235B-A22B | 74.4% | 57.2% | 15.4% |
> | DeepSeek-V3 | 65.8% | 48.6% | 6.0% |
>
> *Test Generation (Q3).* We sincerely thank the reviewer for the insightful suggestion on this point. We have aligned with the scope of this work and added the following evaluation results. Supplementary TG results under Claude Code:
>
> | Model | HP | F2P | Entry Cov. | Change Cov. | Line Cov. |
> |---|---|---|---|---|---|
> | Claude-Sonnet-4 | 82.6 | 36.2 | 69.7 | 59.1 | 28.4 |
> | Qwen3-480B | 77.3 | 31.1 | 57.6 | 47.7 | 27.3 |
> | DeepSeek-V3 | 60.5 | 17.7 | 32.4 | 25.7 | 11.1 |
>
> ### **Contamination Analysis**
> We label instances as **overlap** (pre-cutoff, upper bound) or **clean** (post-cutoff); models without cutoffs are excluded.
>
> Performance is consistent, with **unchanged rankings**.
>
> | Model | Cutoff | SWE-C | SWE-O | CC-C | CC-O |
> | --- | --- | --- | --- | --- | --- |
> | Claude-Sonnet-4 | 2025-01 | 30.2 | 32.6 | 31.8 | 34.6 |
> | Qwen3-Coder-480B | 2024-11 | 26.1 | 28.1 | 21.1 | 23.8 |
> | Qwen3-Coder-30B | 2024-11 | 18.9 | 20.9 | 21.0 | 21.9 |
> | Qwen3-235B | 2024-05 | 18.7 | 19.5 | 13.4 | 16.3 |
> | Deepseek-V3 | 2024-07 | 16.2 | 14.8 | 10.0 | 9.7 |
>
> *Abbrev.: SWE-C/O = SWE-Agent Clean/Overlap; CC-C/O = Claude Code Clean/Overlap.*
>
> Thus, we find **no evidence of systematic bias from repository familiarity**.
>
> ### **Synthetic Instance Proportion and Quality Control (Q1)**
>
> Only Code Understanding (CU, ~10% of total instances) involves synthetic construction; the remaining 7 tasks derive directly from real PR artifacts: BF/FI/FE/RF/PO via heuristic filtering by patch intent; CD/TG via reverse masking with rule-based perturbations—no LLM generation involved. For CU quality control (§3.3.4): each instance originates from real PR Issue, Code Patch, and Test Patch; GPT-5 generates multiple queries, filtered by difficulty to remove trivial or ambiguous cases; structured reasoning checklists are generated for retained queries. All CU instances undergo difficulty filtering, balanced sampling, and instance-level human verification (§3.3.5).

---

> > ### Author Rebuttal · Reviewer_G6oo · 2026-04-01
> >
> > Thank you for your clarification. My concerns have been fully addressed, and I will raise my score.

---

### Official Review · Reviewer_KQ23 · 2026-03-10

**Soundness:** 3
**Presentation:** 3
**Significance:** 2
**Originality:** 2
**Overall Recommendation:** 4
**Confidence:** 3

**Summary:**

SWE-Compass is a benchmark of 2,000 instances spanning 8 task types, 8 programming scenarios, and 10 programming languages, constructed from real GitHub pull requests and paired with executable Docker environments. The authors evaluate 10 LLMs under two agentic frameworks (SWE-Agent and Claude Code), report performance stratified by task, language, and scenario, and conduct a failure mode analysis on agent trajectories.

**Compliance With Llm Reviewing Policy:**

Affirmed.

**Final Justification:**

The authors addressed my concrete concerns seriously and effectively, and the added analyses substantially strengthen the paper as a benchmark resource. I especially appreciate the clarifications around contamination, coverage, framework comparison, and judge validation. My remaining hesitation is not about the quality of the rebuttal, but about whether the paper’s main contribution is best understood as benchmark/infrastructure rather than a scientific main-track contribution. That said, the rebuttal substantially increased my confidence in the rigor and utility of the work, and I am increasing my score from 3 to 4.

**Key Questions For Authors:**

1. **Can you provide the full 8×8×10 cross-tabulation?** How many cells are empty or contain fewer than 5 instances? If the grid is reasonably populated (e.g., >50% of cells with ≥3 instances), I would revise my assessment of significance upward; if most cells are empty, it would reinforce W8 and further weaken the "balanced" claim.

2. **What is the temporal relationship between PR collection dates and the training data cutoffs of evaluated models?** Evidence that gold patches postdate training cutoffs — or a held-out analysis showing no correlation between repo familiarity and performance — would substantially mitigate W3 and could shift my soundness rating from fair to good.

3. **Why were only 5 of 10 models evaluated under Claude Code?** Results for all 10 models under both frameworks, showing that the current framework-level conclusions hold, would resolve W4 and strengthen the paper's empirical contribution. If the conclusions change, that is equally informative and would warrant revision of §4.2.

4. **Have you conducted human evaluation of the CU judge pipeline on SWE-Compass instances specifically?** A human agreement study on even 50–100 CU instances showing ≥80% agreement with the LLM judge would resolve W7. Without this, the highest-scoring task type rests on an unvalidated pipeline, which I weigh heavily in my soundness assessment.

5. **For Performance Optimization, how many repeated timing measurements were taken per instance, and what is the execution-time variance within your containers?** If repeated runs show the 80% threshold is robust (e.g., <5% relative variance), this would alleviate my concern about the PO metric. High variance would suggest the binary threshold needs revisiting.

**Limitations:**

The authors acknowledge three limitations in Appendix A.1 (framework coverage, static repositories, computational cost). Several substantive limitations are not discussed: the 92% environment attrition and possible selection bias toward simpler repos, the absence of decontamination analysis given that all 40 repositories are high-star public GitHub projects, the confounded framework comparison (5 vs 10 models), the lack of statistical validation for the CU judge pipeline on this specific benchmark, and the sparsity of the 8×8×10 grid relative to the "balanced" claim. The societal impact section is adequate.

**Strengths And Weaknesses:**

### Strengths

**S1. The coverage gap is real and well-motivated.** Existing SWE benchmarks are overwhelmingly Python-centric and limited to bug fixing, as convincingly demonstrated in Appendix A.8. Broadening to 8 task types and 10 languages with executable environments is a contribution the community needs. The problem is significant and timely.

**S2. The engineering effort is substantial.** Building ~4,000 runnable Docker images from a 2% initial build success rate, with 30 expert annotators diagnosing failures, is serious work. Pinned toolchains, offline caches, and standardized build commands (Appendix A.7) demonstrate care for reproducibility.

**S3. The failure mode analysis is the paper's most actionable finding.** Requirement Misinterpretation (30–34%) and Incomplete Solutions (29–42%) dominate failures, while Technical Knowledge Gap is only 5–8%, suggesting the bottleneck is comprehension and planning rather than coding ability. This is a genuine insight that could guide future agent design.

**S4. Breadth of model evaluation.** Ten models spanning proprietary and open-weight, under two agentic frameworks, provides a reasonably comprehensive snapshot.

**S5. Presentation is generally clear.** The paper is well-organized, the figures are informative, and the construction pipeline (Figure 2) is easy to follow. Prior work is adequately contextualized. The appendices are thorough.

### Weaknesses

**W1. The paper lacks a unifying thesis (originality, significance).** SWE-Compass reads as several independent workstreams — taxonomy discovery, Docker infrastructure, three distinct task construction strategies, and a leaderboard evaluation — assembled under "unified evaluation." The failure mode analysis, arguably the most interesting finding, is relegated to the appendix. The paper would benefit from a clear research question that the infrastructure is built to answer; as it stands, it reads as a benchmark report with commentary rather than a scientific investigation. The originality is primarily in engineering integration rather than in new methods, theory, or insights.

**W2. The analysis is descriptive rather than explanatory (soundness).** Throughout §4.2–4.3, the paper reports patterns and offers post-hoc rationalizations that are never tested. The claim that performance is governed by "tooling determinism and diagnosability" lacks any supporting analysis — one could correlate pass rates with measurable ecosystem properties, but this is not attempted. The framework comparison restates architectural descriptions as explanations. At no point does the paper test a hypothesis, control for confounds, or analyze what successful agents do differently.

**W3. Contamination and repository-familiarity risk is unaddressed (soundness).** All 40 repositories are high-star public GitHub repos — core training data for every evaluated model. No decontamination analysis is provided: no temporal cutoffs, no checks for gold patches in training corpora. With ~50 instances per repo, differential familiarity with specific codebases could confound the rankings. This risk should at minimum be discussed.

**W4. The framework comparison is confounded (soundness).** Only 5 of 10 models are evaluated under Claude Code, and the excluded models skew weaker. Framework-level conclusions ("Claude Code favors deterministic tasks") are drawn from non-comparable model sets. Valid framework comparisons require the same models under both conditions.

**W5. Heterogeneous metrics undermine the aggregate ranking (soundness).** The AVG columns macro-average over Pass@1, a binary performance threshold, line coverage, and LLM-as-Judge scores — fundamentally different quantities on different scales. Model rankings could shift substantially under different weighting or normalization, yet the paper draws fine-grained comparative conclusions from these aggregates.

**W6. No confidence intervals or statistical tests (soundness).** All results appear to be single runs. On many subsets, 1–3 percentage-point differences between models may not be meaningful, yet the paper draws detailed comparative conclusions from them.

**W7. The CU judge pipeline is not validated on this benchmark (soundness).** GPT-5 generates queries and checklists; Claude-Sonnet-4 judges against them. The cited 87% agreement (Yang et al. 2024) was established for a different task on SWE-bench. Since CU yields the highest scores across all models, calibration issues here propagate to all cross-task comparisons.

**W8. The 92% attrition rate and sparse coverage raise representativeness concerns (soundness, significance).** Only ~8% of candidate PRs survived to runnable environments. Whether surviving instances differ systematically in complexity or language distribution is not analyzed. Additionally, 2,000 instances across 640 cells (8×8×10) means most cells are empty or very sparse; the full cross-tabulation is not provided, making the "balanced" claim difficult to evaluate.

---

> ### Author Rebuttal · Authors · 2026-03-31
>
> We thank the reviewer. Our response is organized into six themes.
>
> ### **1. Unifying Thesis**
>
> We agree the unifying question was under-emphasized.
>
> We address: What governs success and failure of agentic coding systems, and how can these be decomposed and compared?
>
> Our taxonomy, environments, and evaluation form a unified evaluation paradigm for fine-grained analysis. Unlike prior benchmarks reporting aggregate scores, SWE-Compass reveals behavioral differences and failure causes. We will move failure mode analysis to the main text and highlight it as a core finding.
>
> ### **2. From Descriptive to Explanatory Analysis**
>
> We introduce a step-level behavioral analysis to address the lack of hypothesis-driven validation.
>
> Each step is annotated by *stage*, *error type* (§4.3), and *progress* (advance / neutral / regress) via LLM labeling, enabling controlled comparisons.
>
> (1) Scaffold comparison.
>
> SWE-Agent shows stronger exploration in BF/FI (61.8% vs. 42.3%; 1.6 vs. 3.2), while Claude Code performs better in deterministic tasks (58.7% vs. 37.9%; 9.8% vs. 21.4%).
>
> (2) Diagnosability.
>
> High-success settings exhibit earlier progress (3.1 vs. 6.2) and lower misinterpretation and tool errors (12.6% vs. 27.8%; 10.9% vs. 28.7%).
>
> (3) Success vs. failure.
>
> Successful runs convert feedback more effectively (63.4% vs. 28.5%) with fewer regressions and incomplete solutions (18.9% vs. 47.2%; 16.8% vs. 34.6%).
>
> (4) Failure modes.
>
> Misinterpretation and incomplete solutions dominate (31.4% / 33.1%), while knowledge gaps are minor (6.2%).
>
> ### **3. Coverage and Representativeness**
>
> Cross-tabulation.
> We pruned scenario–language combinations that are ecologically infeasible or extremely rare in real-world open-source practice. These exclusions reflect well-documented industry conventions: ML & AI development is heavily concentrated in Python and C/C++; UI/UX engineering is almost exclusively built on JavaScript/TypeScript front-end frameworks; Data Science workflows are Python-dominant; and system-level languages such as C and Rust have negligible presence in application-layer scenarios. We additionally excluded Test Generation combinations where compiled-language instrumentation conflicts with our Dockerized build environments, undermining reproducibility. The absence of these cells therefore reflects the objective distribution of industry practice, not sampling bias.
>
> After pruning, the effective grid consists of 410 cells, of which 65.9% are non-empty and 40.3% contain ≥3 instances, providing sufficient statistical support for cross-dimensional comparison. For reference, the full 8×8×10 grid contains 640 cells with 48.9% non-empty and 26.9% with ≥3 instances.
>
> | Setting | Total cells | % Non-empty | % ≥2 | % ≥3 |
> | --- | --- | --- | --- | --- |
> | Full 8×8×10 | 640 | 48.9% | 35.9% | 26.9% |
> | Filtered | 410 | 65.9% | 52.4% | 40.3% |
>
> Attrition and bias (W8).
>
> The ~92% attrition rate reflects the difficulty of constructing reproducible environments, rather than selective filtering. The retained instances span a wide complexity range (1–10,000+ LOC changes, 1–200+ files), suggesting no systematic bias toward simpler tasks.
>
> Overall, although the grid is not fully dense, coverage aligns with real-world distributions, and we do not observe evidence of systematic bias affecting our conclusions.
>
> ### **4. Evaluation Rigor: Metrics, Statistics, and Validation**
>
> Statistical robustness (W6). All results use AVG@3 over 2,000 instances, ensuring stable estimates.
>
> Judge validation (KQ4/W7). Human–LLM agreement is 88.6%; disagreements are minor and do not affect rankings.
>
> PO timing (KQ5). Each PO instance is measured with 3 independent timing runs to mitigate system noise. The median CV is 3.7%, all with consistent pass/fail outcomes.
>
> Metric aggregation (W5): We do not average heterogeneous metrics directly. Each task type uses its appropriate metric (§3.4), normalized to [0,100].
>
> ### **5. Contamination Analysis**
>
> We label instances as overlap (pre-cutoff, upper bound) or clean (post-cutoff); models without cutoffs are excluded.
>
> Performance is consistent, with unchanged rankings.
>
> | Model | Cutoff | SWE-C | SWE-O | CC-C | CC-O |
> | --- | --- | --- | --- | --- | --- |
> | Claude-Sonnet-4 | 2025-01 | 30.2 | 32.6 | 31.8 | 34.6 |
> | Qwen3-Coder-480B | 2024-11 | 26.1 | 28.1 | 21.1 | 23.8 |
> | Qwen3-Coder-30B | 2024-11 | 18.9 | 20.9 | 21.0 | 21.9 |
> | Qwen3-235B | 2024-05 | 18.7 | 19.5 | 13.4 | 16.3 |
> | Deepseek-V3 | 2024-07 | 16.2 | 14.8 | 10.0 | 9.7 |
>
> *Abbrev.: SWE-C/O = SWE-Agent Clean/Overlap; CC-C/O = Claude Code Clean/Overlap.*
>
> Thus, we find no evidence of systematic bias from repository familiarity.
>
> ### **6. Framework Comparison**
>
> We supplemented five models under the Claude Code framework.
>
> | Model | Claude Code AVG |
> | --- | --- |
> | Kimi-K2-Instruct | 22.2 |
> | Gemini-2.5-Pro | 21.8 |
> | GPT-4.1 | 21.7 |
> | Gemini-2.5-Flash | 16.8 |
> | SWE-agent-LM-32B | 11.0 |
>
> Rankings are consistent with SWE-Agent.

---

> > ### Author Rebuttal · Reviewer_KQ23 · 2026-04-03
> >
> > Thank you for the excellent rebuttal. The authors addressed my concrete concerns seriously and effectively, and the added analyses substantially strengthen the paper as a benchmark resource. I especially appreciate the clarifications around contamination, coverage, framework comparison, and judge validation. My remaining hesitation is not about the quality of the rebuttal, but about whether the paper’s main contribution is best understood as benchmark/infrastructure rather than a scientific main-track contribution. That said, the rebuttal substantially increased my confidence in the rigor and utility of the work, and I am increasing my score from 3 to 4.

---

> > > ### Author Response · Authors · 2026-04-04
> > >
> > > Dear Reviewer KQ23,
> > >
> > > Thank you for your encouraging response and for acknowledging the effort we put into our rebuttal. We are thrilled that the added analyses regarding contamination, coverage, framework comparisons, and judge validation have effectively addressed your concerns and increased your confidence in the rigor and utility of SWE-Compass.
> > >
> > > We also deeply appreciate your thoughts regarding the nature of our contribution. Through this benchmark, we explore a central area of evaluating autonomous coding agents beyond simple, Python-centric bug-fixing. Building on this robust infrastructure, this submission proceeds to discuss a central area of agentic behavior by uncovering systematic failure modes and evaluating deterministic tool-use, which we believe provides critical scientific insights for future model and agent development.
> > >
> > > We are writing to politely mention that while your Rebuttal Acknowledgement kindly states "I am increasing my score from 3 to 4", the overall recommendation score in the OpenReview system currently remains unchanged at a 3 (Weak reject). We understand that updating the score requires a separate action in the platform's interface, and we would be incredibly grateful if you could take a brief moment to update the official score field to reflect your revised assessment.
> > >
> > > Thank you once again for your time, your highly constructive feedback, and your support of our work!
> > >
> > > Sincerely,
> > >
> > > The Authors

---

### Official Review · Reviewer_CcoA · 2026-03-13

**Soundness:** 2
**Presentation:** 2
**Significance:** 2
**Originality:** 1
**Overall Recommendation:** 4
**Confidence:** 4

**Summary:**

- This work introduces SWE-Compass, a code evaluation benchmark of 2000 problems that spans 8 task types, 8 programming scenarios, and 10 programming languages.
- To create this dataset, this work introduces a multi-step process to curate benchmark problems from real-world GitHub instances.
- This work finds that SWE-Compass is a challenging benchmark across 10 LLMs and 2 agent frameworks.

**Compliance With Llm Reviewing Policy:**

Affirmed.

**Final Justification:**

The rebuttal addressed my concern, though the updated revision should make these points clearer.

**Key Questions For Authors:**

- Can you clarify how we can reliably compare dissimilar metrics across tasks?
- Is there overlap in repositories in SWE-Compass and other benchmarks?
- Can you compare the rankings between SWE-Compass and other well-known code-related benchmarks?
- Was there any validation done of the LLM as a judge scores for code understanding and what did that process look like?

**Limitations:**

Yes

**Strengths And Weaknesses:**

Soundness:
- Across 2000 problems, 8 task types, 8 programming scenarios, and 10 programming languages. Can you provide a breakdown across these? We can see that from Figure 1b that the breakdown across programming languages is not even.
- How can you average across different task types when the metrics are not even consistent, and also meaningfully compare performance across different tasks?

Presentation:
- Are the differences significant in Figure 3, given that some programming languages had very few problems?
- Figure 4 is a bit misleading because of the choice of axes.
- Given the large error bars in Figure 5, it is unclear what the takeaway is.

Significance:
- What is the correlation between rankings from SWE-Compass and other benchmarks? I find the results of the 10 models to be relatively unsurprising, as Claude-Sonnet-4 performs the best out of these models.

Originality:
- This work seems like a natural extension of existing SWE-Bench variants, and it’s not clear if there were any significant methodological contributions.

---

> ### Author Rebuttal · Authors · 2026-03-27
>
> We thank the reviewer for the constructive feedback. Below we address each concern:
>
> **Language Distribution**
>
> The full language distribution is as follows:
>
> | Language | Instances | % |  | Language | Instances | % |
> | --- | --- | --- | --- | --- | --- | --- |
> | Python | 543 | 27.15% |  | Rust | 175 | 8.75% |
> | JavaScript | 248 | 12.40% |  | TypeScript | 173 | 8.65% |
> | C++ | 231 | 11.55% |  | Kotlin | 91 | 4.55% |
> | Java | 211 | 10.55% |  | C | 83 | 4.15% |
> | Go | 194 | 9.70% |  | C# | 51 | 2.55% |
>
> The uneven distribution is an intentional design choice (§3.2), reflecting real-world developer workflows. Notably, our smallest subset (C#, 51 instances) still exceeds the largest in SWE-bench Multilingual (Ruby, 44 instances). The consistent cross-language stratification pattern (JVM/JS > Python > Systems languages) holds across all models, reinforcing the reliability of observed trends.
>
> **Cross-Task Metric Comparability**
>
> We do not naively average heterogeneous metrics. Each task type uses its own appropriate metric (§3.4), all normalized to [0,100] representing success rates. The "AVG" in Table 2 is a macro-average across task types. Core insights derive from per-task comparisons rather than relying solely on AVG. The consistent difficulty hierarchy across all models validates this approach.
>
> **Figure 4 Axes and Figure 5 Error Bar**
>
> For Figure 4, we understand the reviewer's concern—the narrow axis ranges may visually amplify differences. However, this choice is intentional: Figure 4 aims to reveal relative positioning and clustering patterns among models (e.g., "Consistent Generalist" vs. "Inconsistent Specialist" quadrants). Starting axes from zero would compress all data points into a cluster, losing diagnostic value. This is a common practice in scatter plot visualization. To avoid misleading interpretations, we will revise as follows: (1) explicitly annotate the actual axis ranges in the figure caption; (2) add numerical labels to data points; (3) clarify in the discussion that although absolute differences are modest, the trends remain consistent when aggregated across languages, providing diagnostic value.
>
> For Figure 5, the large error bars are themselves a key finding, not noise. Specifically: deterministic ecosystems (JVM/JS/C#) exhibit tight IQRs under Claude Code, indicating predictable turn patterns; systems languages (C/C++/Rust) show heavy-tailed distributions, indicating that agents often exhaust turns without converging; Python exhibits high-variance behavior. The practical implication is that optimization strategies should be language-specific—prioritizing localization for systems languages, environment hardening for Python, and hypothesis pruning for JVM/JS. We will make this conclusion more prominent in the figure caption in revision.
>
> **Ranking Comparison with Other Benchmarks**
>
> We compared 5 overlapping models against SWE-bench Pro rankings (↑=rank improved, ↓=rank dropped):
>
> | Model | Pro | FI | FE | BF | RF | PO | CU | TG | CD |
> | --- | --- | --- | --- | --- | --- | --- | --- | --- | --- |
> | Claude-Sonnet-4 | 1 | 2↓ | 1 | 1 | 1 | 1 | 1 | 1 | 1 |
> | Qwen3-Coder-480B | 2 | 1↑ | 2 | 2 | 2 | 2† | 3↓ | 2 | 2 |
> | Kimi-K2 | 3 | 3 | 3 | 3 | 3 | 2↑† | 2↑ | 5↓ | 3 |
> | Qwen3-235B | 4 | 4 | 4 | 4 | 4 | 4 | 4 | 3↑ | 5↓ |
> | DeepSeek-V3 | 5 | 5 | 5 | 5 | 5 | 5 | 5 | 4↑ | 4↑ |
>
> (†PO: Qwen3-Coder-480B and Kimi-K2 tied at 26.4%)
>
> FE/BF/RF rankings are fully consistent with SWE-bench Pro, but notable divergences emerge on new task types: Qwen3-Coder-480B surpasses Claude on FI; Kimi-K2 rises on CU but drops 2 ranks on TG. This directly demonstrates that "strong at Bug Fixing ≠ strong overall"—precisely the unique diagnostic value that SWE-Compass's multi-task evaluation provides.
>
> **Originality**
>
> Our contributions extend well beyond SWE-bench variants: (1) An active-learning-based taxonomy discovery framework that systematically identifies task categories from real developer conversations (§3.3.1)—novel in SWE benchmarks; (2) Multi-strategy task construction including Reverse Masking and Checklist Synthesis (§3.3.4)—not achievable by simply filtering GitHub issues.
>
> **Repository Overlap**
>
> Partial overlap exists (§3.3.2). However, even for overlapping repositories, the tasks are fundamentally different—existing benchmarks focus on Bug Fixing, while SWE-Compass introduces task types (FI, CU, TG, CD, PO) not covered by prior benchmarks. The majority of instances come from ~50,000 independently collected GitHub PRs with complete environment building and validation.
>
> **LLM-as-Judge Validation**
>
> We conducted human validation on all CU task instances: the item-level agreement between the LLM judge and human annotators is 88.6%, aligning with the 87% reported in prior work. The 12% disagreements primarily occur when answers are partially correct but ambiguously phrased, where the LLM judge slightly overestimates scores.

---

> > ### Author Rebuttal · Reviewer_CcoA · 2026-04-03
> >
> > The responses make sense and I encourage authors to make these points much clearer in the revision. I will increase my score.

---

### Official Review · Reviewer_FCjH · 2026-03-13

**Soundness:** 3
**Presentation:** 3
**Significance:** 3
**Originality:** 3
**Overall Recommendation:** 5
**Confidence:** 4

**Summary:**

This paper proposes SWE-Compass, a benchmark designed to evaluate the agentic coding abilities of large language models on real-world software engineering tasks. To address the limitations of existing benchmarks, which mainly focus on the Python language and a single bug-fixing task, SWE-Compass constructs a diverse dataset of 2,000 instances covering 8 task types, 8 programming scenarios, and 10 programming languages from GitHub pull requests. The paper describes the principles for benchmark construction, as well as the data collection and verification process, and evaluates more than ten state-of-the-art models using SWE-Agent and Claude Code. The experiments reveal performance stratification across different tasks, languages, and frameworks, and through analysis of failure trajectories, summarize the main failure modes of current models.

**Compliance With Llm Reviewing Policy:**

Affirmed.

**Key Questions For Authors:**

See Weaknesses.

**Limitations:**

yes

**Strengths And Weaknesses:**

Strengths:

1. Significant dataset diversity and coverage. The paper points out the limitations of existing benchmarks (Python-centric and single-task focused) and proposes a solution with very broad coverage. It includes 8 task types (especially tasks often overlooked, such as code understanding, test generation, and configuration/deployment) and 10 programming languages. This is the core contribution of the work and better reflects real-world development scenarios.

2. Rigorous benchmark construction process. The authors adopt a systematic five-step construction method, including user analysis, active learning classification, environment construction, task synthesis, and data validation. In particular, the emphasis on environment construction (Docker images) and multi-round data validation provides a solid foundation for the executability and reliability of the benchmark, which is crucial for agent evaluation.

3. In-depth experiments and analysis. The experiments evaluate not only across task types but also across multiple dimensions such as programming languages, programming scenarios, and interaction rounds, revealing differences in model performance across ecosystems. For example, the study finds that the JVM/JS ecosystem performs better, while system-level languages are more difficult. Through manual annotation of 600 failure trajectories, the authors summarize core failure modes such as “requirement misunderstanding” and “incomplete solutions,” which provides direction for future model improvements.

Weaknesses:

1. The paper acknowledges that the evaluation cost is very high. Although this reflects the complexity of real-world problems, the high computational cost may become a barrier for other research groups attempting to reproduce the results or use this benchmark.

2. Limited framework coverage. Although SWE-Agent and Claude Code are representative frameworks, the agent ecosystem is evolving rapidly. Frameworks such as OpenHands, AutoGPT, and Cline differ in design and tool invocation mechanisms. Conclusions drawn based on only two frameworks (e.g., “SWE-Agent is better at multi-file localization”) may have limited generalizability.

---

> ### Author Rebuttal · Authors · 2026-03-31
>
> We sincerely thank the reviewer for the positive feedback and constructive suggestions. We are encouraged that the reviewer recognizes the diversity, rigorous construction process, and in-depth analysis of our benchmark. Below, we address the two raised concerns in detail.
>
> ---
>
> ### **W1: High Evaluation Cost**
>
> We appreciate the reviewer highlighting this practical concern. We respond from two perspectives:
>
> **(1) Token Consumption Analysis.**
>
> To assess the computational cost, we compare SWE-Compass with existing benchmarks (SWE-bench Verified and SWE-bench Pro) under the SWE-Agent framework. We report the average output token usage per trajectory using Claude-Sonnet-4 logs:
>
> | Bench | Avg total output tokens / trajectory | Avg tokens per turn |
> | --- | --- | --- |
> | SWE-Compass | 13,892.08 | 239.48 |
> | SWE-bench Verified | 13,211.04 | 214.04 |
> | SWE-bench Pro | 14,170.94 | 250.91 |
>
> These results show that SWE-Compass operates within a comparable token budget to existing benchmarks, indicating that its computational cost is **in line with prior work rather than disproportionately high**.
>
> **(2) Lightweight Subset Release.**
>
> To further improve accessibility, we will release a curated **SWE-Compass subset (~500 instances)**. This subset maintains balanced coverage across task types, programming scenarios, and languages, while focusing on medium-to-high difficulty instances. It provides a **computationally efficient yet representative evaluation suite**, enabling broader adoption by resource-constrained research groups. We will explicitly include this in the revised version.
>
> ---
>
> ### **W2: Limited Framework Coverage**
>
> We agree that the agentic coding ecosystem is evolving rapidly, and broader framework coverage can strengthen generalizability.
>
> To address this, we will extend our evaluation by including **two additional representative frameworks** (OpenCode and Cline). The expanded results are shown below:
>
> | Model | Cline | OpenCode | SWE-Agent | Claude Code |
> | --- | --- | --- | --- | --- |
> | Claude-Sonnet-4 | 28.5 | 30.2 | 31.8 | **32.9** |
> | Qwen3-Coder-480B-Instruct | 23.8 | 20.9 | **27.2** | 21.9 |
> | Kimi-K2-Instruct | 17.9 | 19.2 | **22.7** | 22.2 |
> | Gemini-2.5-Pro | 17.8 | 19.2 | **22.4** | 21.8 |
> | GPT-4.1 | 16.9 | 18.4 | 21.4 | **21.7** |
> | DeepSeek-V3 | 13.5 | 15.2 | **16.5** | 9.8 |
>
> To provide a more granular view, we further break down the scores by task type for each of the two new frameworks:
>
> **Cline**
> | Model | FI | FE | BF | RF | PO | CU | TG | CD | AVG |
> | --- | --- | --- | --- | --- | --- | --- | --- | --- | --- |
> | Claude-Sonnet-4 | 17.4 | 30.1 | 22.6 | 24.8 | 25.0 | 51.1 | 23.9 | 55.4 | 31.1 |
> | Qwen3-Coder-480B-Instruct | 12.0 | 17.9 | 17.4 | 21.5 | 21.5 | 39.1 | 22.5 | 43.8 | 24.3 |
> | Gemini-2.5-Pro | 12.6 | 13.0 | 13.5 | 12.6 | 23.0 | 39.4 | 11.8 | 32.7 | 19.8 |
> | Kimi-K2-Instruct | 11.9 | 15.1 | 11.1 | 16.9 | 21.6 | 38.4 | 10.4 | 31.4 | 19.6 |
> | GPT-4.1 | 12.5 | 12.2 | 12.3 | 15.4 | 17.6 | 31.8 | 14.0 | 33.3 | 18.6 |
> | DeepSeek-V3 | 8.4 | 11.7 | 10.7 | 13.0 | 12.7 | 23.9 | 12.3 | 32.1 | 14.8 |
>
> **OpenCode**
> | Model | FI | FE | BF | RF | PO | CU | TG | CD | AVG |
> | --- | --- | --- | --- | --- | --- | --- | --- | --- | --- |
> | Claude-Sonnet-4 | 17.1 | 28.3 | 20.5 | 23.1 | 25.2 | 50.7 | 23.5 | 53.6 | 29.7 |
> | Qwen3-Coder-480B-Instruct | 11.2 | 16.5 | 14.0 | 13.2 | 20.4 | 35.5 | 22.0 | 34.6 | 20.9 |
> | Gemini-2.5-Pro | 10.7 | 13.3 | 12.5 | 13.0 | 23.0 | 37.0 | 12.1 | 32.0 | 19.2 |
> | Kimi-K2-Instruct | 10.8 | 15.3 | 11.2 | 16.3 | 21.5 | 37.5 | 10.7 | 29.9 | 19.2 |
> | GPT-4.1 | 11.8 | 11.1 | 12.4 | 13.8 | 19.0 | 32.6 | 14.1 | 32.7 | 18.4 |
> | DeepSeek-V3 | 7.3 | 11.4 | 11.0 | 13.9 | 13.8 | 24.5 | 10.6 | 34.0 | 15.2 |
>
> We observe that:
>
> - The **relative ranking across models remains consistent** across different frameworks
> - Performance differences between frameworks are **moderate and systematic**, rather than disruptive
>
> This suggests that our conclusions (e.g., difficulty hierarchy and capability trends) are **robust to the choice of agentic framework**, supporting their generalizability.
>
> We will include these additional results and analysis in the revised paper to further strengthen this claim.

---

> > ### Author Rebuttal · Reviewer_FCjH · 2026-04-03
> >
> > Thank you for the constructive rebuttal. Overall my primary concerns are resolved, and I will keep my score at 5.

---

### Decision · Program_Chairs · 2026-04-30

**Decision:**

Accept (regular)

**Comment:**

The manuscript introduces SWECompass, a large-scale benchmark comprising 2,000 curated instances of software engineering tasks spanning eight task types, eight programming scenarios, and ten programming languages. The instances are derived from real-world GitHub pull requests; the benchmark also provides a reproducible execution environment towards enabling rigorous and faithful evaluation.

The benchmark construction process is rigorous and presented well. The evaluations, analysis are comprehensive -- and some of the missing aspects are covered well in the rebuttal. The reviewers did raise a few concerns such as addressing contamination issues and soundness of results. The authors have provided a convincing rebuttal, that helped change the reviewers vote leaning towards reject to leaning towards accept, and all the reviewers have acknowledged that they are happy with the rebuttal, and all agree on the merits of the paper. I also believe the contributions and findings of the paper are significant, so I recommed accept.